



# Model simulations of atmospheric methane and their evaluation using AGAGE/NOAA surface- and IAGOS-CARIBIC aircraft observations, 1997-2014

Peter H. Zimmermann[1], Carl A. M. Brenninkmeijer[1], Andrea Pozzer[1], Patrick Jöckel[3], Andreas Zahn[2], Sander Houweling[4], and Jos Lelieveld[1]

[1] Max Planck Institute for Chemistry, Department of Atmospheric Chemistry, Mainz, Germany
[2] Karlsruhe Institute of Technology (KIT), Institute for Meteorology and Climate Research, Karlsruhe, Germany
[3] Deutsches Zentrum für Luft- und Raumfahrt (DLR), Institut für Physik der Atmosphaere, Oberpfaffenhofen, Germany
[4] Netherlands Institute for Space Research, Utrecht, the Netherlands

*Correspondence to*: Peter H. Zimmermann (p.zimmermann@mpic.de)

**Abstract.**

The global budget and trends of atmospheric methane ($CH_4$) have been simulated with the EMAC atmospheric chemistry – general circulation model for the period 1997 through 2014. Observations from AGAGE and NOAA surface stations and intercontinental CARIBIC flights indicate a transient period of declining methane increase during 1997 through 1999, followed by seven years of stagnation and a sudden resumed increase after 2006. Starting the simulation with a global methane distribution, scaled to match the station measurements in January 1997 and using inter-annually constant $CH_4$ sources from eleven categories together with photochemical and soil sinks, the model reproduces the observations during the transient and constant period from 1997 through 2006 in magnitude as well as seasonal and synoptic variability.

The atmospheric $CH_4$ calculations in our model setup are linearly dependent on the source strengths, allowing source segregated simulation of eleven biogenic and fossil emission categories (tagging), with the aim to analyze global observations and derive the source specific $CH_4$ steady state lifetimes. Moreover, tagging enables a-posteriori rescaling of individual emissions with proportional effects on the corresponding inventories and offers a method to approximate the station measurements in terms of lowest RMS. Enhancing the a priori biogenic tropical wetland emissions by ~29 Tg/y, compensated by a reduction of anthropogenic fossil $CH_4$ emissions, the all-station mean dry air mole fraction of 1792 nmol/mol could be simulated within a RMS of 0.37 %. The coefficient of determination $R^2 = 0.87$ indicates good agreement with observed variability and the calculated 2000-2005 average interhemispheric methane difference between selected NH and SH stations of 119 nmol/mol matches the observations.

The $CH_4$ samples from 95 intercontinental CARIBIC flights for the period 1997-2006 are also accurately simulated by the model, with a 2000-2006 average $CH_4$ mixing ratio of 1786 nmol/mol, and 65 % of the measured variability being captured. This includes tropospheric and stratospheric data. To explain the growth of $CH_4$ from 2007 through 2013 in term of sources, an emission increase of 28.3 Tg/y $CH_4$ is needed. We explore the contributions of two potential causes, one representing natural emissions from wetlands in the tropics and the other anthropogenic shale gas production emissions in North America. A 62.6 % tropical wetland contribution and of 37.4 % by shale gas emissions optimally fit the trend, and simulates $CH_4$ from 2007 – 2013 with an RMS of 7.1 nmol/mol (0.39 %). The coefficient of determination of $R^2 = 0.91$ indicates even higher significance than before 2006. The 4287 samples collected during 232 CARIBIC flights after 2007 are simulated with an RMS of





1.3 % and $R^2$ = 0.8, indicating that the model reproduces the seasonal and synoptic variability of $CH_4$ in the upper troposphere and lower stratosphere.

## 1 Introduction

The greenhouse gas methane ($CH_4$) is emitted into the atmosphere by various natural and anthropogenic sources, and is removed by photochemical reactions and to a small extent by soils. The tropospheric mean lifetime of $CH_4$

due to oxidation by OH has been estimated to be 8-9 years (Lelieveld et al., 2016) and its concentration has been growing by about 1 %/y since the beginning of the Anthropocene in the 19[th] century (Crutzen, 2002).

The resulting 2.5 fold increase of the global $CH_4$ abundance since pre-industrial times produces a climate forcing of 0.57 $Wm^{-2}$ (direct 0.44$Wm^{-2}$, indirect 0.13W $m^{-2}$) which is about 35 % of the climate forcing by $CO_2$ (1.6$Wm^{-2}$) (Lelieveld et al., 1998; Dlugokencky et al., 2011). In the IPCC fifth report an even higher forcing of

0.97 $Wm^{-2}$ corresponding to about 57 % of the climate forcing by $CO_2$ (1.68$Wm^{-2}$) is assessed (IPCC, 2013). After the strong upward $CH_4$ trend since the 1960s, by the end of the 1990s the increase had slowed down until sources and sinks quasi balanced for about 8 years, while in 2007 the $CH_4$ increase resumed unexpectedly (Bergamaschi et al., 2013). Fig. 1 demonstrates the development of the $CH_4$-mixing ratio at the AGAGE observation site Cape Grim, Australia (41° S, 145º E) over the years 1997 through 2014, the period considered in

this modeling study and reveal a no-trend period from 2000 through 2006.

The resuming upward trend after 2007 is not fully understood: data analysis (Nisbet et al., 2016) and inverse modelling studies (Bergamaschi et al., 2013) indicate that global emissions since 2007 were about 15 to 22 Tg $CH_4$/y higher than in previous years, possibly caused by increasing tropical wetland emissions and anthropogenic pollution in mid-latitudes of the northern hemisphere. A potentially growing source that was

identified is hydraulic shale gas fracturing, for instance in Utah where 6 to 12 % of the natural gas produced may locally leak to the atmosphere (Karion et al., 2013, Helmig et al. 2016). The increasing production of fossil fuels to some extend may explain the $CH_4$ trend; however Schaefer at al. (2016) by means of $^{13}C/^{12}C(CH_4)$ data and a box model concluded that fossil fuel related emissions are a minor contributor the renewed methane increase. Simultaneously, "since 2007 $\delta^{13}C$-$CH_4$ (a measure of the $^{13}C/^{12}C$ isotope ratio in methane) has shifted to

significantly more negative values suggesting that the methane rise was dominated by significant increases in biogenic methane emissions, particularly in the tropics, for example, from expansion of tropical wetlands in years with strongly positive rainfall anomalies or emissions from increased agricultural sources such as ruminants and rice paddies " (Nisbet et al., 2016).

The causes of the trend changes have been subject of a number of studies, some with contradictory results (for

instance, Simpson et al., 2012, and Kai et al., 2012) highlighting the complexity of the processes that control the methane budget during this part of the Anthropocene, combined with a paucity of data.

As mentioned above, Schaefer et al. (2016) showed that "after 2006, the activation of biogenic emissions caused the renewed $CH_4$ rise", raising concern about the contribution from rice production versus wetland emissions, and Schwietzke et al. (2016), based on reassessment of data of the $^{13}C/^{12}C$ ratio of $CH_4$ from fossil sources,

conclude that the assumed global fossil fuel $CH_4$ emissions need a major upward revision of 60-110 %. In other words, it was found that the combined fossil $CH_4$ sources (1985-2002) must have been much stronger (factor of 2), at the expense of microbial sources. Further, it was concluded that fossil fuel related sources had decreased. Although the findings of the two articles are not necessarily in conflict, their results warrant further work on the



methane budget. Hausmann et al. (2016), using methane and ethane column measurements, concluded that the

increase in $CH_4$ since 2007 has been for 18 to 73 % (depending on assumed ethane/methane source ratios) due to thermogenic methane. Further, Helmig et al. (2016) suggested a large contribution of US oil and natural gas production to the increased emissions. Saunois et al. (2016), in an extensive review of the methane budget, conclude that $CH_4$ emissions from agricultural activities seem to be a major, possibly dominant cause of the atmospheric growth trend of the past decade. This not only puts the focus on biogenic versus thermogenic, but

highlights that both source types are directly influenced by human activity, with the option of being controlled by the implementation of policies.

Here we investigate how well, based on source estimates, $CH_4$ concentrations and their changes over the past two decades can be simulated numerically, by accounting for atmospheric dynamical and chemical processes with the atmospheric chemistry-general circulation model EMAC, which describes the transport, dispersion, and

chemistry of atmospheric trace constituents, and allows the online sampling of calculated mixing ratios in four dimensions, mimicking the sampling by observational systems (Jöckel et al., 2010). To evaluate the simulation results we use $CH_4$ concentration data at surface stations, i.e. data from AGAGE (Prinn et al., 2000) and NOAA (Dlugokencky et al., 2016) and $CH_4$ data collected by the CARIBIC passenger aircraft (Brenninkmeijer et al., 2007).

Both measurement data sets (i.e. the surface station based and the aircraft based) allow a global approach, with each having its characteristic "footprint". The station data are based on regular measurements at fixed coordinates in both hemispheres. The CARIBIC data (Civil Aircraft for the Regular observation of the atmosphere Based on an Instrumented Container) are based on monthly flight series (nominally 4 sequential long-distance flights) covering large parts of the globe from a Eurocentric perspective.

**2 Model Setup**

**2.1 The EMAC numerical model**

The ECHAM/MESSy Atmospheric Chemistry (EMAC) model is a chemistry and climate simulation system that includes sub-models describing tropospheric and middle atmosphere processes and their interaction with oceans, land and human influences. The Modular Earth Submodel System (MESSy, www.messy-interface.org) results

from an open, multi-institutional project providing a strategy for developing comprehensive Earth System Models (ESMs) with flexible levels of complexity. MESSy describes atmospheric chemistry and meteorological processes in a modular framework, following strict coding standards. The sub-models in EMAC have been coupled to the 5th generation European Centre HAMburg general circulation model (ECHAM5, Röckner et al., 2006), of which the coding has been optimized for this purpose (Jöckel et al, 2006, 2010).

The extended EMAC model version 2.50 at T106L90MA resolution was used to simulate the global methane budget. A triangular truncation at wave number 106 for the spectral core of ECHAM5 corresponds to a (~1.1°×1.1°) horizontal quadratic Gaussian grid spacing near the equator, and 90 levels on a hybrid-pressure grid in the vertical direction span from the Earth's surface to 0.01 hPa pressure altitude (~80km, the middle of uppermost layer). The vertical resolution near the tropopause is about 500 m. Numerical stability criteria require

an integration time step of 1-2 min. With regard to model dynamics, we applied a weak "nudging" towards realistic meteorology over the period of interest, more specifically by Newtonian relaxation of four prognostic



model variables temperature, divergence, vorticity and the logarithm of surface pressure towards operational analysis data of the European Centre for Medium-range Weather Forecasting (ECMWF) (van Aalst et al., 2004). Apart from the prescribed sea surface temperature (SST) and the nudged surface pressure, the nudging method is

applied in the free troposphere, tapering off towards the surface and tropopause, so that stratospheric dynamics are calculated freely, and possible inconsistencies between the boundary layer representations of the ECMWF and ECHAM models are avoided. Further, in the free troposphere, the nudging is weak enough to not disturb the self-consistent model physics, while this approach allows a direct comparison of the model output with measurement data (without constraining the model physics), and therefore offers an efficient model evaluation.

The EMAC sub-model collection includes "CH4" which is tailored for stratospheric and tropospheric methane chemistry and solves the ordinary differential equations describing the oxidation of methane by OH, $O^1D$, Cl and photolysis. The feedback to the hydrological cycle by modification of the specific humidity is optional in $CH_4$ and switched off in this particular setup for the same reason as applying tropospheric nudging as mentioned above. The water that is produced by methane oxidation is in the used setup not added to the hydrological cycle.

This is indeed only relevant in the stratosphere.

As long as the tracers under consideration are not subject to chemical feedback reactions among each other, they can be processed separately. In this manner, atmospheric constituents such as methane can be tagged e.g. by the source category which they derive from and simulated individually, while their sum exactly fits the simultaneous total $CH_4$ calculations. In our particular case, no feedback is affecting the prescribed OH distribution neither in

the gross nor in the tagged mode. (cf CH. 2.2.3).

The sub-models "TIMEPOS" and "S4D" enable online sampling of model parameters such as tracer mixing ratio at selected observation sites as well as along aircraft measuring flight routes (http://www.messy-interface.org/ "MESSy Submodels" and Jöckel et al., 2010).

### 2.2 Methane sources and sinks

### 2.2.1 Methane emissions

The combined input from eleven inter-annually constant natural and anthropogenic methane source types amounts to 580 TgCH₄/y, applied to the simulation period 1997 – 2014.

Anthropogenic and natural methane sources are based on The Global Atmospheric Methane Synthesis (GAMeS), a GAIM/IGBP (http://gaim.unh.edu/) initiative to develop a process-based understanding of the

global atmospheric methane budget for use in predicting future atmospheric methane burdens. Emission data for this initiative have been used for the model setup described here. Natural wetland emissions are based on Walter et al. (2000) and Fung et al. (1991). Processes with similar isotopic characteristics are aggregated into one group. Oil related sources, for example, comprise mining and processing of crude fuel and all emission classes related to the use of fossil fuel such as residential heating, on/offshore traffic, industry, etc., and also includes an

estimate of volcanoes (Houweling et al., 1999). Given that methane emissions from boreal/arctic wetlands are quite uncertain, it is reasonable to assume that this source category accounts for permafrost decomposition emissions as well.

The "burning"-part of the GAMeS dataset is replaced by the GFED statistics (van der Werf et al., 2017) in addition to biofuel combustion emissions from the EDGAR2.0 database (Olivier, 2001). The biogenic emissions

from bogs, rice fields, swamps and biomass burning are subject to seasonal variability (Tab. 1). About 60 % of the total emissions of 580 Tg/y are caused by human activities; the remainder is from natural sources. At





northern middle and high latitudes, methane sources predominantly comprise animals (ruminants), bogs, gas and coal production, transmission and use, landfills, and boreal biomass fires. Tropical wetlands (partly in the subtropics) are the world's largest (natural) source of methane together with animals. Minor tropical

anthropogenic input is from biofuel combustion. The individual source strengths are partly subject to seasonal variability, and except for yearly differences in the ~20 Tg/y biomass burning, are assumed to be inter-annually constant in a reference simulation for the full period 1997 through 2014. Fig. 2 depicts the total emission distribution ($gCH_4$ $m^{-2}$ $month^{-1}$) for Jan. (a) and Jul. (b), in logarithmic scale for better representation, to illustrate seasonal $CH_4$ changes.

A rearrangement among the natural wetland and the anthropogenic landfill-, coal-, gas-, and oil contributions by 30 $Tg(CH_4)/y$ (i.e. 5 % of the total) in favor of the low latitude wetlands has been evaluated retrospectively under the condition of least RMS deviation between station and model $CH_4$ mixing ratios (Table 1, column 2).

The horizontal resolution of all methane fluxes is 1°×1°. Because biomass burning emissions are associated with thermal uplift, they are vertically distributed up to 3000 m and higher according to a profile suggested in

EDGAR3.2ft (Aardenne et al., 2005). The GFED biomass burning statistics include agricultural waste burning events.

Additional emission sources are necessary to close the budget during the methane rising period after 2006. The contributions by enhanced release from tropical wetlands (TRO) and North American shale gas drilling (SHA) (FracFocus, 2016) are discussed in Sect. 4.2.

### 175 2.2.2 Methane uptake by soils

A small but significant (6.6 % in this study) removal process of methane is its oxidation by methanotrophic bacteria in soils (Dlugokencky et al., 2010). In absence of a well-defined deposition frequency, a negative mission flux of 37.8 Tg ($CH_4$)/y dependent on season (e.g., 2.4 Tg in January and 4.0 Tg in July) at the surface is applied, based on Ridgwell et al. (1999). The negative flux distribution has a pronounced seasonal cycle in phase

with the emissions and depends on soil temperature, moisture content and the land cultivation fraction.

### 2.2.3 Methane chemical removal

The chemical removal process of $CH_4$ is photo-oxidation, predominantly by hydroxyl (OH) radicals. In addition to the reaction with OH in the troposphere and stratosphere, there are minor oxidation reactions with atomic chlorine (Cl) in the marine boundary layer and the stratosphere and with electronically excited oxygen atoms

($O(^1D)$) in the stratosphere (Lelieveld et al., 1998; Dlugokencky et al., 2011). In EMAC the methane photolysis and chemical reaction system is numerically solved by the sub-model "CH4". Global distributions of OH, Cl, and $O(^1D)$ have been pre-calculated from the model evaluation reference simulation S1 (Jöckel et al., 2006), therefore providing self-consistent oxidation fields for the model transport and chemistry of precursors. Monthly averaged fields calculated for the year 2000 have been used in this study.

### 190 3 Observations used for model verification

The EMAC model simulates the global distribution of methane from given emission source categories, and produces time series of methane distributions as output. Additionally, model samples during the simulation are recorded for the verification of the results at prescribed locations and times. Monthly averaged mixing ratios are



computed at the location of selected AGAGE/NOAA sites and about 4600 CARIBIC flight measuring samples
(Brenninkmeijer et al., 1999, 2007) gathered during more than 350 flights from 1997 through 2014. The station
records predominantly serve as a reference for the model evaluation and help to gain confidence in the CARIBIC
flight data analysis and interpretation.

### 3.1 AGAGE and NOAA station network

The ALE/GAGE/AGAGE stations are coastal and mountain sites around the world chosen primarily to provide
accurate measurements of trace gases with lifetimes that are long compared to global atmospheric circulation
times (Prinn et al., 1978, 2013). The AGAGE sites used in this study are (Fig. S1): Cape Grim, Australia (41º S,
145º E), Cape Mata Tula, American Samoa (14º S, 171º W), Mace Head, Ireland (53º N, 10º W), Ragged Point,
Barbados (13º N, 59º W) and Trinidad Head, California (41º N, 124º W), and the NOAA site Mauna Loa,
Hawaii, in the United States (19.5°N, 155.6°W, 3397 masl) (Dlugokencky et al., 2015). In the following we refer
to the stations as CGO, SMO, MHD, RPB, THD and MLO, respectively. Monthly mean mixing - unfiltered with
respect to local pollution events - are compared to respective monthly averaged model samples stored every day
at 12:00 GMT.

### 3.2 CARIBIC flight observations

CARIBIC (Civil Aircraft for the Regular Investigation of the Atmosphere Based on an Instrument Container,
Brenninkmeijer et al., 2007) is a European, passenger aircraft based atmospheric composition monitoring project
that has become part of the IAGOS Infrastructure (www.iagos.org). CARIBIC deploys an airfreight container
equipped with about 1.5 tons of instruments, connected to a multi-probe air inlet system. The container is
installed monthly for 4 sequential measurement flights from and back to Frankfurt Airport after which air
samples, aerosol samples and data are retrieved. The container houses instruments for measuring ozone, carbon
monoxide, nitrogen oxides, water vapor and many more trace gases as well as atmospheric aerosols. Air samples
are collected at cruise altitudes between about 10 and 12 km and depending on latitude and season and actual
synoptic meteorological conditions represent tropospheric or stratospheric air masses.

Overall the ratio between sampled stratospheric and tropospheric air masses is about 0.5. These air samples are
analyzed in the laboratories of the CARIBIC partner community. More than 40 gases are measured including
hydrocarbons, halocarbons and greenhouse gases including $CH_4$. Methane mixing ratios were determined at
coordinates along flight tracks over regions such as Europe (EUR), North America (NAM), South America –
north (SAN), South America – south (SAS), Africa (AFR), India and Indonesia (IND), and Far East (FAE) and
color coded in Fig. S2. These values, interpolated in time and space onto the model grid, are subject of our
evaluation.
For further information about CARIBIC based studies involving $CH_4$, we refer to Schuck et al. 2012, Baker et al.
2012, and Rauthe-Schöch et al. 2016. For the period 1997-2002, we use data from the first phase of CARIBIC
(Brenninkmeijer et al. 1999).

### 4 Simulation results

Starting with a global distribution derived from spin-up simulations and scaled to match the 1997 station
measurements, a time series of the monthly mean global methane distribution up to December 2014 has been





calculated together with daily online samples at AGAGE/NOAA stations and along the CARIBIC flight tracks for comparison. Characteristic features, such as global $CH_4$ distributions and seasonal cycles as well as the local variability of station and flight records can be successfully reproduced for the first three years 1997 – 1999, during the slowing increase, as well as the subsequent period through 2006 without a trend.

In our specific model setup, the oxidation chemistry, neglecting $H_2O$-feedback, and parameterized soil removal process of $CH_4$ respond linearly to the emissions, thus allowing the separate simulation of individual sources by tagging. Consequently, the sum of eleven tagged methane simulations exactly reproduces the reference $CH_4$ distribution, and the composition of methane at any grid point in the atmosphere can be attributed to the specific source categories. Taking advantage of the tagging approach, emission sensitivity studies have been performed

in order to approximate the ground station observation series within a least square root deviation (RMS).

Relative to the a priori emissions (Table 1, column 2) a 30 Tg $CH_4$ /y reduction of predominantly northern hemispheric anthropogenic fossil emissions in favor of an 31 Tg/y increment in natural emissions from tropical wetlands results in a minimum all-station-RMS, and accurately reproduces the observed interhemispheric difference between the most northerly and southerly stations.

For the trend period since 2007 we introduced additional emissions, to account for the recent $CH_4$ increase (Kirschke et al., 2013, Miller et al., 2013, Nisbet et al., 2016, Turner et al., 2016). Two hypothetical methane emission scenarios were considered with the aim to explain the discrepancy between observations and the reference simulation from 2007 to 2014: TRO, an additional release from tropical wetlands and SHA, additional emission from North America based on shale gas drilling statistics. Also for this period the least RMS (station

measurement – model simulation) deviation is used as a criterion to evaluate the emission scenarios, together with the slopes of the linear regression trends. A linear optimization analysis guides the attribution of a proportionally larger fraction to TRO than to SHA.

### 4.1 The period 1997 through 2006

For initialization, a global methane distribution pattern for January was created iteratively in several spin-up

cycles and finally rescaled to Jan. 1997 station measurement data. Because of mass conservation, a realistic initial distribution is important to simulate an annual average global $CH_4$ mass that is in steady state over the entire period with inter-annually constant sources and sinks. According to prescribed 4-dimensional coordinate tables, calculated $CH_4$ mixing ratios are recorded and stored at all sampling positions and -times at selected AGAGE (Prinn et al., 2013) and NOAA (Dlugokencky, 2015) observation sites and along the CARIBIC flight

tracks (Brenninkmeijer et al., 1999, 2007) for the years 1997 through 2014 for further graphical and statistical evaluation. Additionally, for the entire time period from 1997 through 2014, based on the mass conserving sources in the EMAC model simulation, a series of global $CH_4$-distributions was produced and stored in 2-day frequency for statistical and graphical evaluation, together with daily (12:00 GMT) model samples at AGAGE/NOAA stations and along the CARIBIC aircraft flight tracks.

The linear dependency between source strength and atmospheric abundance in this specific model setup ensures that the sum of all tagged simulations is equal to the reference simulation comprising the sum of all emissions. Moreover, this numerical property of the model's partial differential equation system allows the redistribution of certain amounts among – e.g. northern and southern - emitters without affecting the global budget, up to minor effects caused by the sink distribution.



While the global total $CH_4$ emissions are relatively well-constrained, estimates of emissions by source category range within a factor of two (Dlugokencky et al. 2011). The global observational networks have shown to be very helpful to derive the emissions at large scales. The CARIBIC observatory provides an additional global view on $CH_4$ abundance and variability in the UTLS, not directly affected by emission sources at the surface, while being sensitive to the vertical exchange of air masses between the lower and upper troposphere.

The use of tagged tracers helps to determine the origin of the methane that is sampled. Tagged initial distributions and tagged soil sinks are calculated as ratios between the respective source fluxes and the total. Corresponding source-segregated $CH_4$ station and aircraft samples were calculated the same way as in the reference simulation, but in this case by each category. Chemical reactions and photolysis are the same as in the simulation with total $CH_4$, i.e. the tagged emissions are exposed to the same oxidant environment. Assuming that

the sources are inter-annually constant, apart from the variability in the comparably small (3.4 %) biomass burning source, the partial masses of the tagged results remain in steady state over the simulation period at roughly proportional amounts to the emission fluxes. However, the exact weighting factors, in terms of the steady-state atmospheric lifetimes, vary somewhat around the integral lifetime of 8.44 years (Fig. 3) because of different exposure to the major chemical destruction areas. The individual steady state lifetimes are quantified in

Ch. 4.1.1.

The integrated model $CH_4$ masses exactly match the mass calculated in the reference simulation with all sources, which confirms the linearity of the system with chemical feedbacks suppressed through the fixed oxidant distributions. Seasonal global mass variations in individual contributions from constant sources are caused by OH-chemistry and dynamics, e.g. by the migrating ITCZ.

**4.1.1 AGAGE/NOAA stations**

Based on the a priori emission assumptions (Tab. 1, col. 2) the 1997 – 2006 average $CH_4$ mixing-ratio over all AGAGE/NOAA stations of 1789 nmol/mol is simulated within a root mean square deviation (RMS) of 0.51 %. With the applied initial distribution (scaled to match the observations) and emissions, the model reproduces both the 1997-1999 trend and the period without trend from 2000-2006. This suggests that the global $CH_4$

concentration in the period 2000-2006 represents the steady state after previously increasing emissions, probably until the early 1990s.

Consistently with the observations, the simulated $CH_4$ mixing ratios are largest at MHD (53°N) and decrease with latitude, reaching a minimum at CGO (41°S). The 2000-2006 (no-trend period) average observed mean mixing ratios for these stations range from 1864 to 1734 nmol/mol and could be reproduced within an average

percentage RMS of 0.88 % and 0.37 % respectively, however, being too high by 12 nmol/mol (0.64 %) at MHD and too low by 5 nmol/mol (0.29 %) at CGO, indicating a possible mismatch in the emission assumptions, which is reflected in an excessive interhemispheric $CH_4$-gradient ΔNS. If we define ΔNS as the difference between average CH4 mixing-ratios at the northern stations MHD (53°N) and THD (41ºN) and the southern station CGO (41ºS) for the years 2000 – 2006, the model-ΔNS of 135 nmol/mol appears too large compared to the observed

118 nmol/mol. Although this imparity could also be caused by erroneous interhemispheric transport, previous analyses (Aghedo et al.2010, Krol et al. 2017) show that the underlying ECHAM5 model reproduce realistically the Inter-hemispheric transport time.

The tagging approach offers a way to rescale emission amounts of individual sources with proportional effects on the global distribution. An emission increment in the tropical wetland source (SWA) along with a reduction



of nearly the same amount of the fossil group of categories comprising landfill-, coal-, gas-, and oil (FOS) appears to be appropriate to improve the latitudinal $CH_4$ distribution. The model to observation RMS deviation as a function of a FOS to SWA redistribution amount follows a $2^{nd}$ order polynomial shape with minimum RMS = 0.37 % at 24.7 Tg/y (Fig. 4a – blue line). The corresponding ΔNS dependency follows a linear function with root at 30.6 Tg/y (Fig. 4a – red line). The range between the optima in RMS and ΔNS is a consequence of the

relatively course discrimination of the contributing sources into SWA and just FOS. The average redistribution value of 28.7 Tg/y in form of a RMS/ΔNS combined optimum is applied in the following (Table 1, col. 3), especially as a basis for the 2007 – 2014 methane trend simulation. The average mixing ratios at the stations in latitudinal order are plotted in Fig. 4b.

In Fig. 5 the observed $CH_4$ records at our reference stations 1997 - 2006 are plotted in blue together with the

standard deviations and compared to model results (red) gained with the a posteriori emissions. The seasonality, with a maximum in NH-winter, is apparent at MHD, but is more pronounced and less scattered at THD, RPB and MLO. Near the Equator (e.g. SMO, 14° S) the seasonal cycle is weakest, while at CGO in the SH we find a phase shift by six months compared to NH stations. Regression analysis (Fig. 6) shows that the coefficient of determination is highest at CGO ($R^2$ = 0.92) but still highly significant at MHD (0.84), despite that the station is

relatively close to methane sources and relatively frequently affected by synoptic scale pollution events from the European continent. Please refer to Table 2, row C4 for $R^2$ and row C3 for the RMS at all stations.

At MLO (3397masl) the offset of -287 nmol/mol together with the slope of 1.16x indicates an underestimation of low $CH_4$-values there and can be attributed to increasing vertical model grid resolution with altitude; i.e. the amplitude is reduced but the average preserved.

The tagged tracer simulations indicate that the atmospheric concentrations during the stable phase 1997 through 2006 are proportional to the respective emission amounts, but influenced by the distance from the source due to the oxidation by OH (Fig. 7). Emissions that take place relatively close to the main sinks (i.e. predominantly tropical OH) have a reduced atmospheric abundance relative to the source strength and vice versa. Particularly boreal biomass burning emissions with accumulating concentrations over the NH (Fig. S3a) have an extended

lifetime of 8.92 years, compared with rice paddy emissions, which are subject to injection into the tropical global OH maximum (Fig. S3b) and relatively short-lived (8.34 y). All stations are more than 25 % exposed to tropical and southern hemispheric swamp-released methane, ranging from 25.8 % at MHD, 53°N to 28.9 % at CGO, 40°S. Landfill emissions, for example, are less intense than those of swamps and predominantly released in the NH. Their footprint at stations undergoes an opposite (north/south) gradient (Fig. 8).

From 2007 on, when the station records show an upward trend (cf. Fig. 1 representatively for CGO) additional emissions are necessary in order to close the budget if the sink processes are kept unchanged. The simulation for this period is presented in Ch. 4.2.

**4.1.2 CARIBIC flights**

The spatio-temporal distribution of the CARIBIC $CH_4$ sampling is quite different from that of the surface

stations. Measurements are taken over relatively short time intervals and more than 96 % of the samples are from the NH. In contrast to the monthly average station data, the CARIBIC individual methane observations in the Upper Troposphere and Lower Stratosphere (UTLS) are based on air sampling over 20 minutes (i.e. ~300 km) for CARIBIC-1 and about two minutes (i.e. ~30 km) for CARIBIC-2 and compared to the stations appear to be much more variable. The sequence of sampling is irregular in time, i.e. the same destinations are reached through





different flight routes (Fig. S2), and take place at different times of the year. Thus the following statistics are not comparable to the station observations.

Between 2000 and 2006, the average of all simulated methane samples based on the a posteriori emission data is 1781 nmol/mol while the corresponding CARIBIC observations average at 1786 nmol/mol with a RMS deviation of 1.05 % and a coefficient of determination $R^2 = 0.65$ (Table 3, rows C1-4). The scattered sampling

positions cannot be accurately reproduced by the grid model EMAC, because of its limited resolution. The observed $CH_4$ variability features short-duration events like the interception of methane plumes or alternatively relatively clean air episodes and even stratospheric air, however, the patterns are rather well reproduced (Fig. 9). The model appears to capture the variations well, even those which are subject to intercepting upper tropospheric and lowermost stratosphere at mid and higher latitudes.

The amplitudes of the model time series, however, are smaller due to the relatively coarse vertical grid spacing of the model, which represents the UTLS at a vertical resolution of about 500m – compared to ~45m near surface. In contrast to background station measurements, for the CARIBIC time series local maxima and minima are not only related to season but also to vertical gradient effects, especially due to the strong concentration changes across the tropopause. The scatter plot (Fig. 10) shows a regression slope of 0.54, i.e. well below 1,

which quantifies the evident underestimation of the calculated $CH_4$ variability in the graphs of Fig. 9, suggesting that the vertical resolution of the model grid is not optimal to resolve the fine structure in the tropopause region. The slope is compensated by a corresponding offset, up to 817 nmol/mol, explaining the good agreement between simulations and observations in Fig. 9.

For further analysis, according to the definition in Sect. 3.2 (Fig. S3), we grouped the data records in Fig. S4 by

the 7 flight sampling regions EUR, AFR, FAE, IND, NAM, SAN, and SAS. The best agreement between model and observations in terms of RMS is achieved over low-latitude regions such as IND with 0.75 % and SAN 0.87 %, while the observations over continental areas in the mid latitude NH still could be simulated within a RMS range of 1.20 % (EUR) and 1.34 % (FAE). It appears that the variance of the CARIBIC measurements is most accurately reproduced over EUR with $R^2 = 0.74$ and over IND with $R^2 = 0.65$. AFR is not discussed here

because of the sparse number of samples of 5.9 %. The statistics are summarized in (Table 3, rows C1-5).

**4.2 Simulating the recent methane trend**

In accordance with the CGO background station data (Fig. 11a, blue line), the CARIBIC measurements show a significant methane increase from 2007 onward (Fig. 11b, blue line), which the model (red) cannot reproduce under the assumption of constant emissions. This discrepancy is removed by assuming an additional inter-

annually constant $CH_4$ source starting in 2007.

Encouraged by our tagging results, an EMAC model sensitivity study was set up with additional emissions from the tropical wetlands (scenario TRO) and North American shale gas (scenario SHA) drilling sites, to resolve the post-2006 model – observation discrepancy mentioned above. Enhanced precipitation in the regional summer season (Nisbet et al., 2016; Bergamaschi et al., 2013) may be a possible cause of higher tropical wetland

emissions. To create a "fracking" map we relied on the publicly available database maintained by the national hydraulic fracturing chemical registry (FracFocus, 2016). Figs. 12a and b show the global $CH_4$ mixing ratios near the surface, logarithmically scaled for better visibility, marking the respective hypothetical emissions. While the assumed Amazon and N-American emission fluxes are identical, the former are more efficiently



mixed vertically due to deep tropical convection (Figs. S5a, b) and therefore lead to smaller enhancements near the surface.

For the amount of additional emissions, we used the upper estimate from Bergamaschi et al. (2013) of 22 Tg CH$_4$/yr as a first guess of emissions to be added in order to fit the upward trend. The resulting slope of CH$_4$ increase over the years 2007 to 2014 at the station CGO turned out to be underestimated by a factor of 1.3, motivating a further increase in emissions. Optimal agreement at CGO was achieved by adding a total of 28.3 Tg/y and was used as input for tagged simulations of scenarios TRO and SHA. Both scenarios perfectly reproduce the observed CH$_4$ trend, but affect the RMS deviation at NH, SH, and tropical stations in different ways. To further optimize the agreement between model and measurements combinations of both scenarios have been evaluated.

Note that in this work we focus on the sources and neglect inter-annual changes in global OH, which are assumed to be small (Nisbet et al., 2016). Changes in the removal rate of methane by the OH radical have not been seen in other tracers of atmospheric chemistry, e.g. methyl chloroform (CH$_3$CCl$_3$) (Montzka et al., 2011; Lelieveld et al. 2016) and do not appear to explain short-term variations in methane. Changes in the order of 3-5% per year over an 8 year period are very unlikely.

In the next sections, more detailed analyses are presented to evaluate the two scenarios.

### 4.2.1 AGAGE/NOAA stations

Shale gas associated emissions, originating mostly from the northern hemisphere, need a relatively longer time to influence CH$_4$ at southern hemispheric stations like Cape Grim, Tasmania (CGO), compared with to those from the tropical wetlands. The latter appear to also affect northern hemispheric observations e.g. at Mace Head, Ireland (MHD). Under the influence of deep convection in the tropics and subsequent global transport, the characteristic seasonality of tropical wetland emissions can significantly influence the CH$_4$ time series worldwide. We use the model results together with the measurement data to estimate to which extend presumed increases in these tropical and extratropical CH$_4$ sources can provide a plausible explanation for the observed recent trend.

As mentioned before, after introducing an emission increment of 28.3 TgCH$_4$/y starting in 2007 for matching the global growth of CH$_4$, the observed trend is reproduced well at all ground stations in both scenarios (TRO and SHA). Similarly to the no-trend period (see Fig. 4a), the relative contribution of TRO and SHA has been obtained by minimizing the ΔNS and RMS with respect to the observations (Fig. 13a, solid lines). The optimized contribution is 62.7 % (17.7 Tg/y) for TRO and 37.3 % (10.6 Tg/y) for SHA. With respect to longitudinal dependency of the SHA emissions, two control simulations were initiated, one with additional emissions from East Asia (FAE: 25° N – 50° N, 100° E - 150° E) and another with additional emissions from Europe (EUR: 45°N – 60°N, 0° - 26°E). The optimization procedure (see Fig.13a, dashed and dotted line for FAE and EUR simulations, respectively) reveals different source fractions for the minima of ΔNS and RMS (see Fig.13a, colored symbols and arrows). This shows that, when the SHA emissions are located away from the North America, no fraction is found that could minimize simultaneously the ΔNS and RMS. On other hand, the discrepancy between the minima of ΔNS and RMS for the scenario with SHA emissions over North America (marked by yellow collate symbols) is very small (within 1%), indicating a realistic latitudinal positioning of the source region.





The scatter plots for the North American reference SHA (Fig. 14) indicate fairly good correlation between the observed vs. calculated station monthly means for the period after 2007. Statistics for the trend simulations are given in Table 2, rows T1-5, including the averaged observed and calculated mixing ratios, their deviation in terms of RMS (%) and the coefficient of determination $R^2$. All trend simulation results at AGAGE/NOAA stations are summarized in Fig. 15.

### 4.2.2 CARIBIC flights

Based on the same optimized emission scenario (62.6 % TRO/ 37.4 % SHA) the trend in the post-2006 CARIBIC-2 methane measurements appears to be realistically simulated by the EMAC model as well. In Fig. 16 monthly averaged CARIBIC measurements are plotted together with corresponding model results. The slopes of the linear trend lines 0.32x (CARIBIC) and 0.31x (EMAC) over time in months over the 8 flight observation years 2007 through 2014, indicating a very good model representation of the methane trend. The regression analysis with $R^2 = 0.8$ (Fig. S7, upper left panel) over all flight samples even improves for this period, possibly also due to a much higher sampling density. As mentioned before the model underestimates the measured extremes, especially negative peaks observed during northern hemispheric intercontinental flights in April and May 2009, '11, and '12 caused by tropopause folds, which at the given vertical grid spacing (~500 m in the respective altitude region) cannot satisfactorily be resolved by the model. The frequency spectrums (Fig.17) confirm this: median simulated values reveal higher amplitudes than measurements before and during the methane-trend period. The different widths of the frequency distributions σ = 6.2 (EMAC) and 4.7 nmol/mol (CARIBIC) for the period 2007-2014 and σ = 7.4 and 6.3 nmol/mol respectively for the period 2000-2006 confirms the model biasing medium range values.

For detailed comparison with the pre-2007 results Fig S6a depicts the whole series unresolved however on non-equidistant time axis. Focusing on individual flight sampling regions (Fig S6b) we restrict the statistical analyses to those areas having at least 300 samples. The highest coefficients of determination ($R^2 > 0.8$) are obtained for NAN, EUR and the FAE. For the other four regions reaching further south such as SAN or IND, the tropopause influence is stronger, leading to reduced linear slopes together with comparably less $R^2$ of 0.58 and 0.72 (cf. Fig. S7 for diagrams).

Individual flights show variations in $CH_4$ source composition in response to relatively small scale influences. A striking demonstration of the varying influences of emissions in the model in regions crossed by the CARIBIC aircraft is provided by flights 244-245 on August 13–14, 2008, between Frankfurt in Germany and Chennai (formerly Madras) in India. In Fig. 18a (right ordinate) the total observed $CH_4$ mixing ratios along the flight track are plotted over the respective simulations with and without TRO/SHA-trend increment. Like at all flights simulated peak values for this flight are underestimated and not correctly in phase with the observations. Fig. 18b underlines this for the whole collection of India bound CARIBIC flight samples in accordance with Fig 17. The TRO/SHA increment in Fig. 18a is obvious but with 0.8 % on average still relatively small in 2008. The source segregated rice paddy-methane (green, left ordinate) dominates the pattern of the total $CH_4$ and the $R^2 = 0.58$ implies that 0.58 % of the observed $CH_4$ variability along this flight track can be explained by rice paddy emissions. Highest mixing ratios in excess of 1850 nmol/mol were recorded in the upper troposphere between 50° and 75° E. Trajectory calculations as well as methane isotope and other chemical tracer analyses (Schuck et al. 2012, Baker et al. 2012) corroborate that these air masses carry emissions from South and Southeast Asia and can be explained by the trapping of air masses (Rauthe-Schöch et al. 2016) from South Asia in the Upper



Troposphere Anticyclone (UTAC), a persistent phenomenon during the monsoon and centered over Pakistan and northern India (Garny and Randel, 2013). This is also qualitatively illustrated in Fig. 19b: The upper

tropospheric trapped regional rice paddy released methane obviously marks the local maximum in the total $CH_4$ distribution (Fig. 19a - different scales were used for better representation). The flight route crosses this pattern twice, from NW to SE and back. Further, relatively localized maxima in the northern hemispheric extra-tropics (red areas in Fig. 19a) are caused by anthropogenic sources such as coal mining and gas exploitation and from the high latitude bogs in summer.

Another demonstrative example for tagging results is presented in Fig. 20 which depicts $CH_4$ mixing-ratios observed during the Far East flight 304 from Osaka, Japan to Frankfurt (Main), Germany in July 2010 together with respective simulations including four of the most relevant individual tagged source contributions. Calculations (red dashed, right axis) follow the phase of the measurements (blue dashed, right axis). The trend period increment (the difference between red fat and red thin lines) in 2010 with 1.44 % in average has almost

doubled compared to 2008.   The pattern is obviously determined by animal-, landfill-, and gas source contributions. The determination coefficients with respect to the observations amount to $R^2 = 0.83$, 0.78, and 0.66 respectively. The pronounced bog-methane profile (color coded in olive-green) is not correctly in phase with CARIBIC in terms of an $R^2 = 0.38$. Rice fields east of 136°E contribute above average.

The systematic study of the source segregated composition of all 327 CARIBIC flights over the years 1997

through 2014 with special emphasis on the developing trend beyond is subject of a separate investigation.

## 5 Conclusion and Outlook

We analyzed the atmospheric methane budget by means of the EMAC model and comparing simulations with data from AGAGE/NOAA surface stations and CARIBIC aircraft data. Source tagging is used to analyze the

emission distribution and to optimize the model results with respect to the observations. We found that, compared to our a-priory assumptions, a larger Amazon wetland source with a concomitant reduction in NH fossil emissions is required to explain the measurements and especially the observed interhemispheric gradient.

Two possible additional methane sources, shale gas extraction (SHA) and tropical wetlands (TRO) have been investigated, that could cause the resuming methane growth since 2007.  We showed that a methane increase of

28.3 Tg/y in 2007 and subsequent years, of which 60 % from TRO and 40 % from SHA, can optimally explain the recent $CH_4$ trend.

In view of the additional global $CH_4$ source since 2007, a source – sink equilibrium has not yet established after the 8 years of emissions considered. A $2^{nd}$ order polynomial extrapolation predicts steady state after 13 years, assuming that the emissions remain unchanged. We are aware that there is no unique solution for the source –

receptor relationship. Therefore, the emissions used in this work must be considered as representative of latitudinal emissions, not representing specific locations of the emissions. Nevertheless, the degree of freedom in the choice of sources is limited and our scenario realistically represents the north-south gradient of $CH_4$, a critical constraint as corroborated by the "Far East" and "Europe" control simulations.

AGAGE/NOAA station methane data are currently available through September 2015 and further updates are

expected. CARIBIC flight measurements are analyzed and data are available through February 2016. We plan to continue the study of these data, supported with EMAC model simulations, also taking advantage of the



resuming CARIBIC flights in 2017/18. A larger coverage of Southern Hemispheric sampling routes would be desirable to extend the database and help explain the ongoing upward methane trend.




**Acknowledgements:**

CARIBIC relevant activities of this modeling project were carried out under contract 320/20585908/IMK-ASF-TOP/GFB by Karlsruhe Institute of Technology (KIT), Karlsruhe.

AGAGE is supported principally by NASA (USA) grants to MIT and SIO, and also by: DECC (UK) and NOAA (USA) grants to Bristol University; CSIRO and BoM (Australia): FOEN grants to Empa (Switzerland); NILU (Norway); SNU (Korea); CMA (China); NIES (Japan); and Urbino University (Italy)





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





**Acronyms:**


| | |
|---|---|
| EMAC | ECHAM/MESSy Atmospheric Chemistry (EMAC) model |
| AGAGE | Advanced Global Atmospheric Gases Experiment |
| NOAA | National Oceanic and Atmospheric Administration |


| | |
|---|---|
| MHD | Mace Head, Ireland (53° N, 10° W) |
| THD | Trinidad Head, California (41° N, 124° W) |
| MLO | Mauna Loa, Hawaii, in the United States (19.5°N, 155.6°W, 3397 masl) |
| RPB | Ragged Point, Barbados (13° N, 59° W) |
| SMO | Cape Mata Tula, American Samoa (14° S, 171° W) |
| CGO | Cape Grim, Australia (41° S, 145° E) |


| | |
|---|---|
| CARIBIC | Civil Aircraft for the Regular observation of the atmosphere Based on an Instrumented Container |
| AFR | Africa |
| EUR | Europe |
| FAE | Far East |
| IND | India |
| NAM | North America |
| SAN | South America north |
| SAS | South America south |



| | |
|---|---|
| TRO | Tropical wetland methane emissions scenario 2007-2014 |
| SHA | Shale gas production methane emissions scenario 2007-2014 |




**Tables**

| CH₄ sources | emission Tg (CH₄)/y | | Seasonality |
|---|---|---|---|
| category | a priory | final[3] | |
| swamps | 133 | 162 | yes |
| animals | 98 | 98 | |
| landfills | 68 | 58 | |
| rice paddies | 60 | 60 | yes |
| gas production | 48 | 41 | |
| bogs | 42 | 42 | yes |
| coal mining | 42 | 36 | |
| oceans + offshore traffic [1] | 17 | | |
| oil production, processing [1] | 8 | | |
| other anthrop. Sources [1,2] | 6 | | |
| volcanoes [1] | 4 | | |
| oil | 35 | 30 | |
| biomass burning | 20 | 20 | yes |
| termites | 19 | 19 | |
| biofuel combustion | 15 | 15 | |
| **sum** | **579** | **580** | yes |

**Table 1:** Methane emissions for EMAC model input 1997 – 2006 (no-trend period).
[1] merged in one category "oil"
[2] all EDGAR emission classes related to the use of fossil fuels such as residential heating, onshore traffic, industry, etc.
[3] redistribution with respect to minimal station observation to model simulation RMS.






| Station: | Mace Head, Ireland | Trinidad Head, CA | Ragged Point, Barbados | Mauna Loa, Hawaii | American Samoa | Cape Grim, Tasmania | Globe |
|---|---|---|---|---|---|---|---|
| Acronym: | MHD | THD | RBP | MLO | SMO | CGO | ALL |
| **No-trend period mean 2000-2005:** | | | | | | | |
| **C1** observations | 1.864E-06 | 1.841E-06 | 1.791E-06 | 1.788E-06 | 1.734E-06 | 1.734E-06 | 1.792E-06 |
| **C2** model | 1.865E-06 | 1.841E-06 | 1.785E-06 | 1.782E-06 | 1.730E-06 | 1.734E-06 | 1.789E-06 |
| **C3** RMS % | 0.51 | 0.36 | 0.36 | 0.46 | 0.33 | 0.19 | 0.37 |
| **C4** $R^2$ | 0.83 | 0.83 | 0.89 | 0.87 | | 0.92 | 0.87 |
| **Trend phase mean 2007-2014:** | | | | | | | |
| **T1** observations | 1.890E-06 | 1.868E-06 | 1.818E-06 | 1.817E-06 | 1.761E-06 | 1.757E-06 | 1.818E-06 |
| **T2** model | 1.896E-06 | 1.867E-06 | 1.814E-06 | 1.811E-06 | 1.758E-06 | 1.761E-06 | 1.818E-06 |
| **T3** RMS % | 0.61 | 0.42 | 0.35 | 0.43 | 0.31 | 0.27 | 0.40 |
| **T4** $R^2$ | 0.89 | 0.86 | 0.91 | 0.93 | 0.89 | 0.97 | 0.91 |

**Table 2:** Statistical evaluation of AGAGE/NOAA ground station methane samples versus EMAC model simulations using optimized emissions.


| Flight region: | Europe | Africa | Far East | India | North America | South Am. north | South Am. South | Globe |
|---|---|---|---|---|---|---|---|---|
| Acronym: | EUR | AFR | FAE | IND | NAM | SAN | SAS | ALL |
| **No-trend period mean 1997-2006:** | | | | | | | | |
| **C1** observations | 1.783E-06 | 1.781E-06 | 1.793E-06 | 1.788E-06 | no flights | 1.786E-06 | 1.778E-06 | 1.786E-06 |
| **C2** model | 1.782E-06 | 1.778E-06 | 1.785E-06 | 1.786E-06 | | 1.778E-06 | 1.773E-06 | 1.781E-06 |
| **C3** RMS % | 1.20 | 0.71 | 1.34 | 0.75 | | 0.87 | 0.84 | 1.05 |
| **C4** $R^2$ | 0.74 | 0.39 | 0.62 | 0.65 | | 0.61 | 0.60 | 0.65 |
| **C5** samples % | 19.60 | 5.86 | 26.63 | 17.92 | 0.00 | 13.07 | 16.92 | 100.00 |
| **Trend phase mean 2007-2014:** | | | | | | | | |
| **T1** observations | 1.791E-06 | 1.802E-06 | 1.802E-06 | 1.811E-06 | 1.773E-06 | 1.813E-06 | 1.839E-06 | 1.801E-06 |
| **T2** model | 1.794E-06 | 1.798E-06 | 1.803E-06 | 1.804E-06 | 1.784E-06 | 1.804E-06 | 1.821E-06 | 1.799E-06 |
| **T3** RMS % | 1.40 | 1.05 | 1.44 | 1.08 | 1.72 | 1.05 | 1.29 | 1.31 |
| **T4** $R^2$ | 0.84 | 0.57 | 0.80 | 0.70 | 0.84 | 0.59 | 0.30 | 0.80 |
| **T5** samples % | 25.7 | 6.9 | 20.4 | 8.7 | 10.3 | 24.6 | 3.5 | 100 |

**Table 3:** Statistical evaluation of CARIBIC flight methane samples versus EMAC model simulations using optimized emissions.






**Figures**

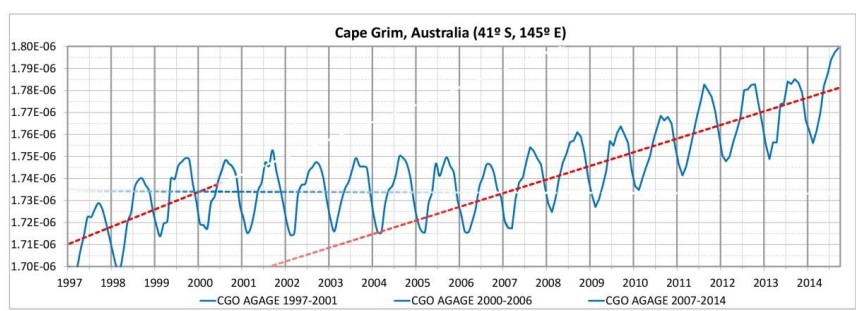

Fig. 1: Development of monthly mean CH$_4$-mixing ratios at the AGAGE observation site Cape Grim, Australia

(41° S, 145° E) over the years 1997 through 2014, the period considered in this modeling study.




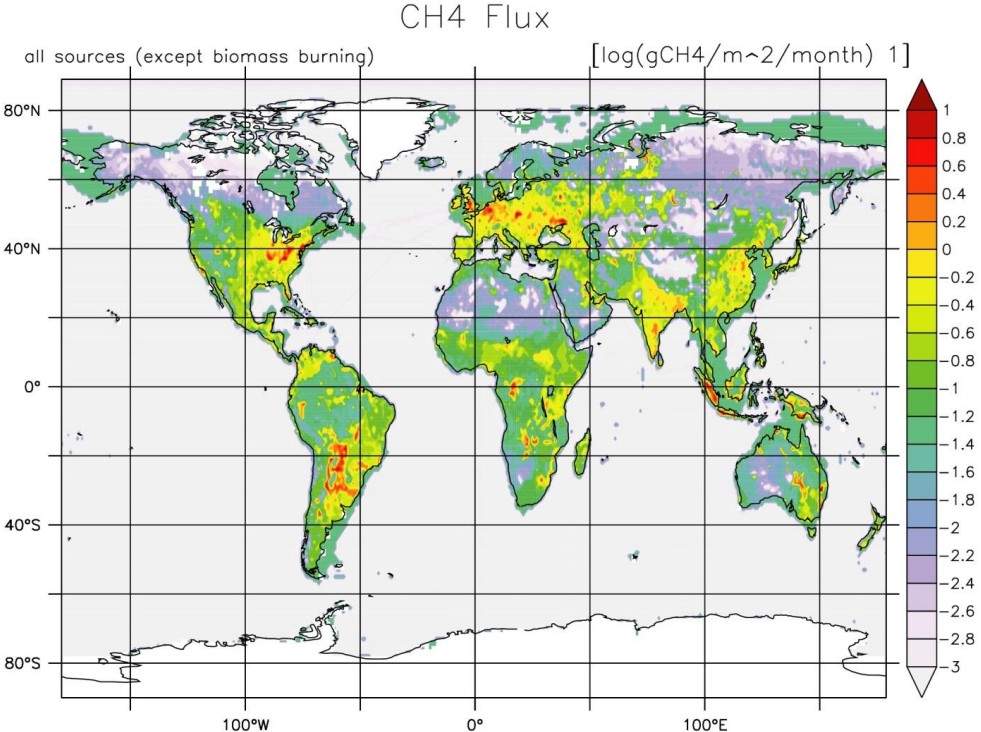

Figure 2:

a: Methane emission flux distribution in January - log (gCH$_4$ m$^{-2}$ month$^{-1}$).





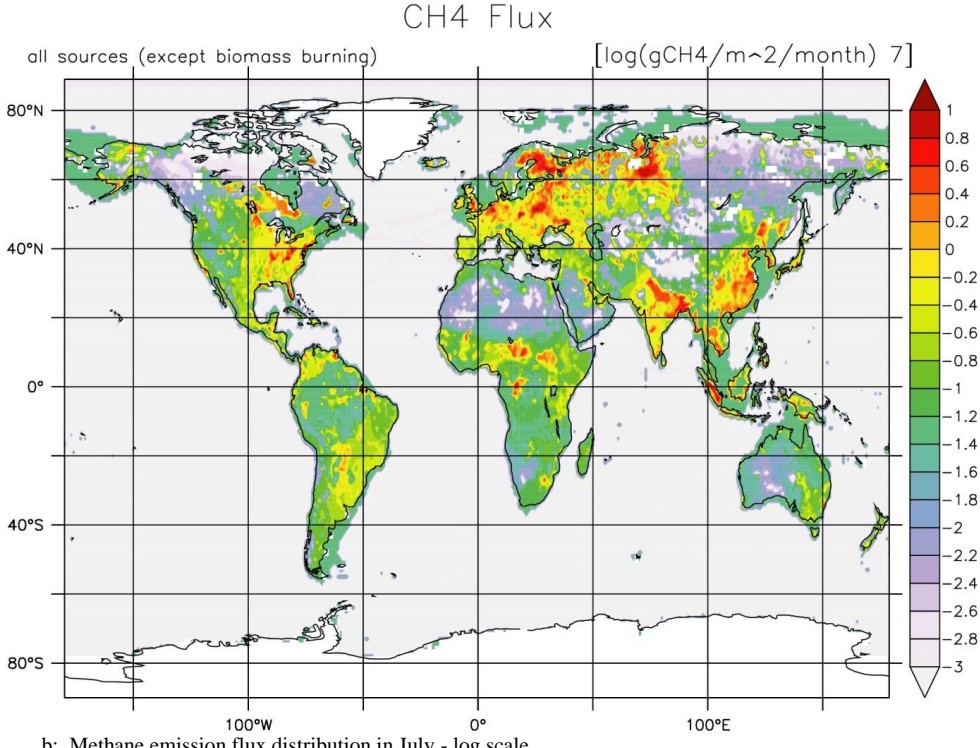

b: Methane emission flux distribution in July - log scale.





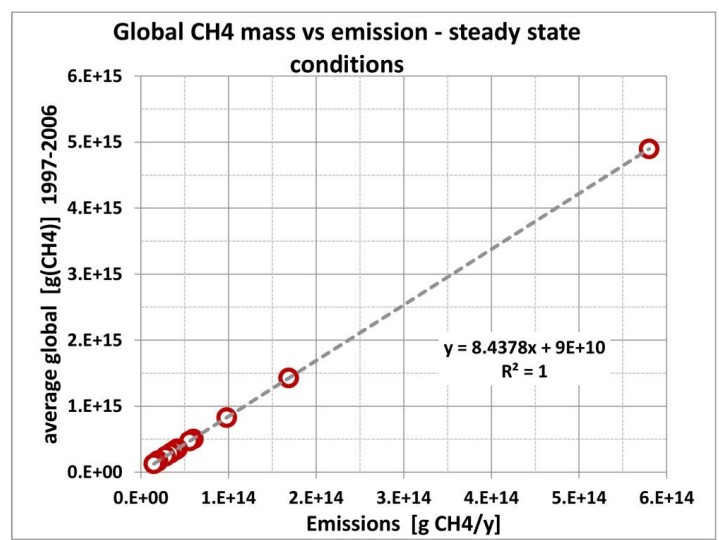

Figure 3: Steady-state global methane mass vs emission amount dependency under steady state conditions.

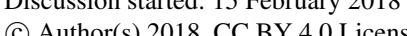



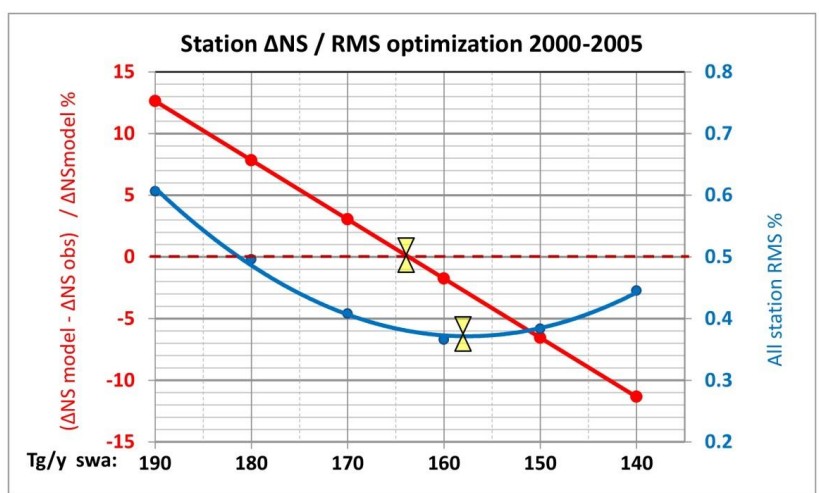

Figure 4:

a.    RMS and ΔNS changes by variating the total CH4 emissions between zero (a priori) and 50 Tg/ yr for
the SWA emissions.





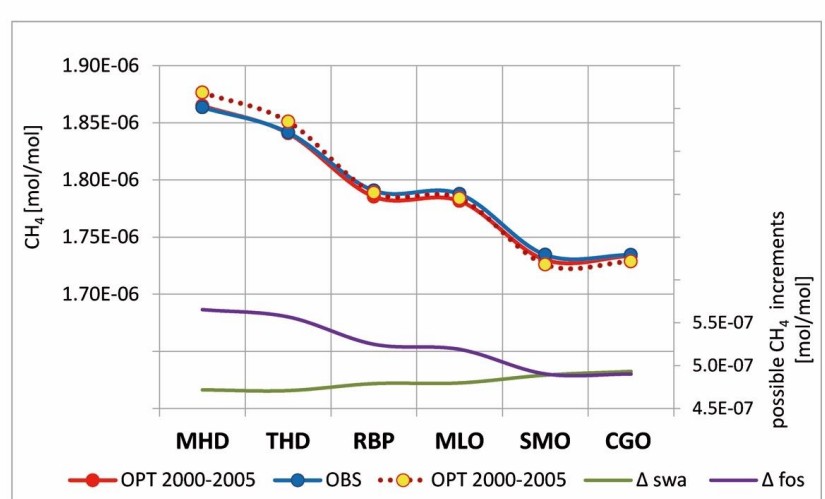

b.  Measured average 2000-2005 methane mixing ratios at the station (blue dots) compared to the a priori
simulation (yellow dots) and the simulation with optimized SWA emissions (red dots). The stations are
ordered by latitude.  The lower lines show the wetland (green) and fossil (purple) net changes necessary for
the optimization





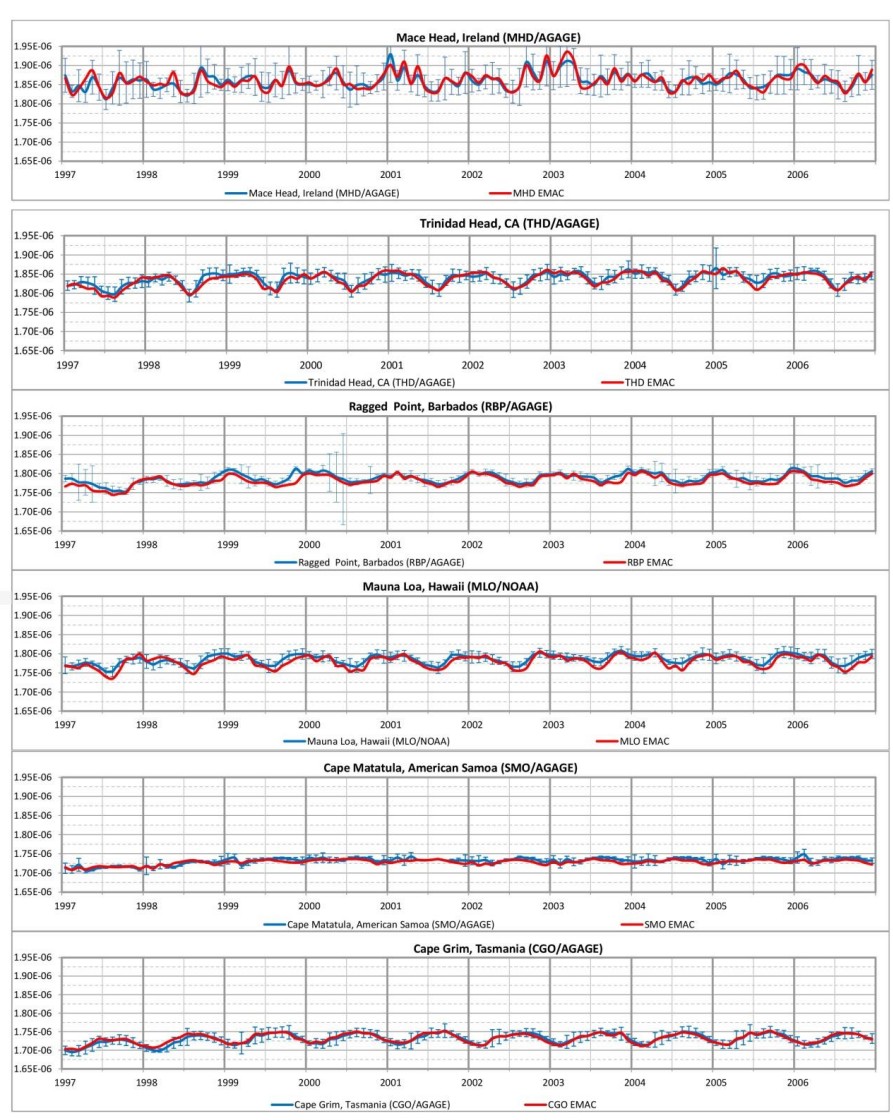

Figure 5: EMAC calculations and AGAGE/NOAA observations of CH$_4$ from 1997 through 2006. The dashed

line shows simulation results without CH$_4$ emission trend.



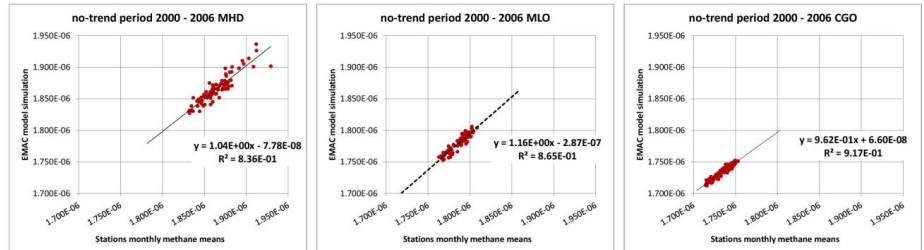

Figure 6: Regression analysis of EMAC calculations vs. observations of CH$_4$ at AGAGE/NOAA stations MHD,
MLO, and CGO for no-trend years 2000 through 2006.

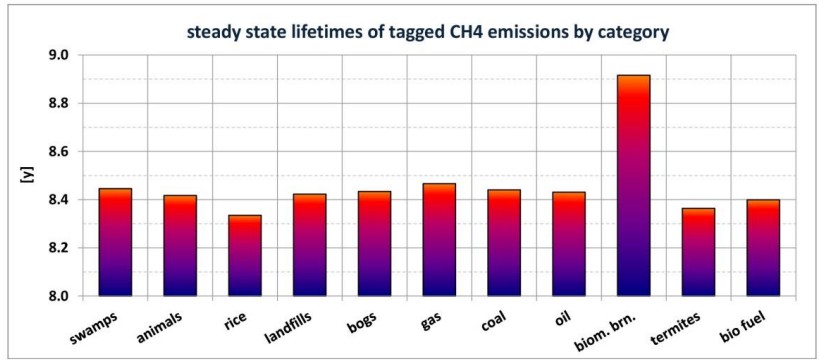

Figure 7: Steady-state atmospheric lifetimes of tagged methane source contributions, 1997 - 2006 (no-trend
period).



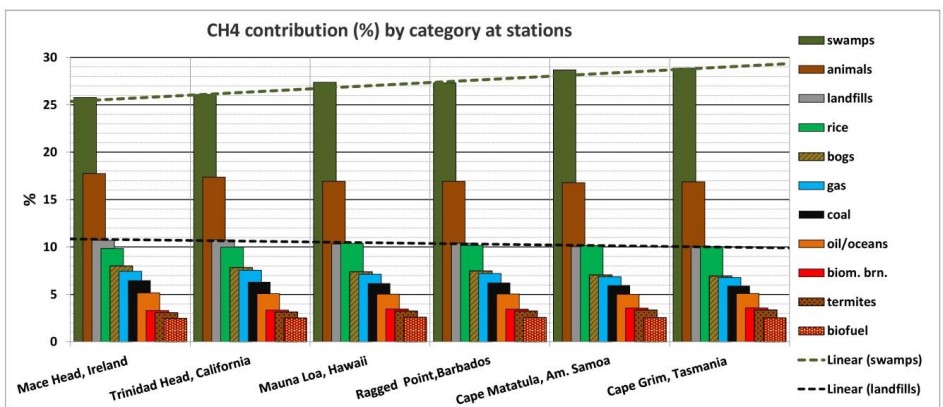


Figure 8: Atmospheric methane composition at AGAGE/NOAA stations (from north to south) during the no-
trend period 1997 - 2006.

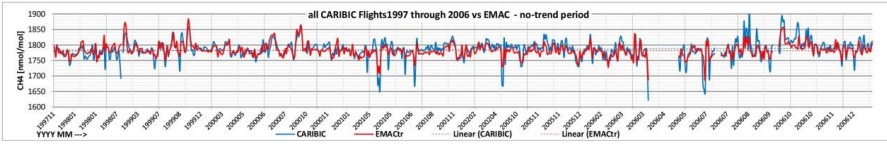


Figure 9: EMAC CH$_4$ calculations (red) and CARIBIC-1/2 observations (blue) from 1997 through 2006,
all flights .




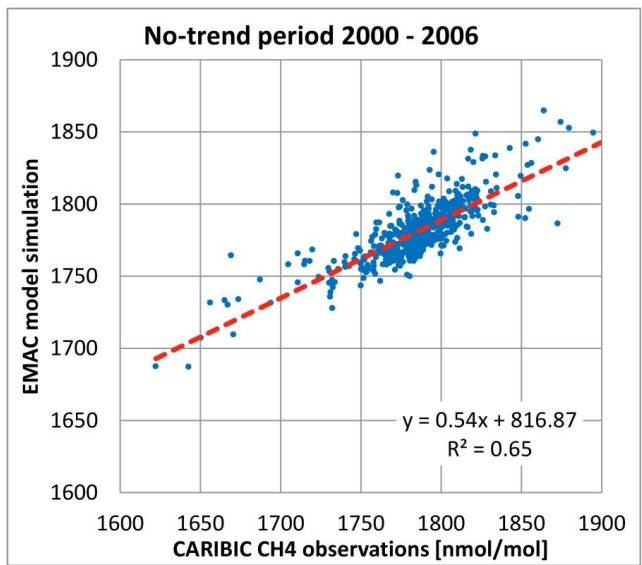


Figure 10:   Correlation EMAC vs. CARIBIC flights 1997 - 2006 (no-trend period).




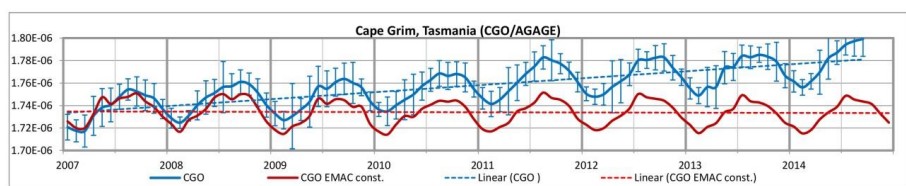

Figure 11:

a: AGAGE observations at CGO (blue) from 2006 through 2014 compared to EMAC CH4 calculations (red) under 1997-2006 unchanged emission assumptions.

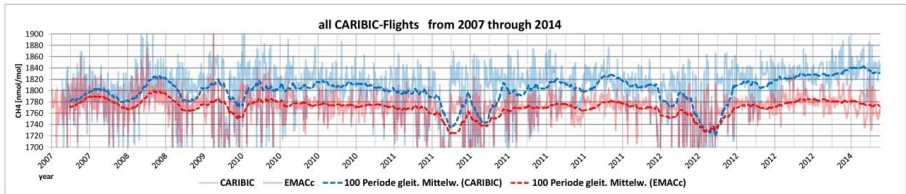

b: same for CARIBIC flights superimposed by respective 100-sample running means for better appearance.





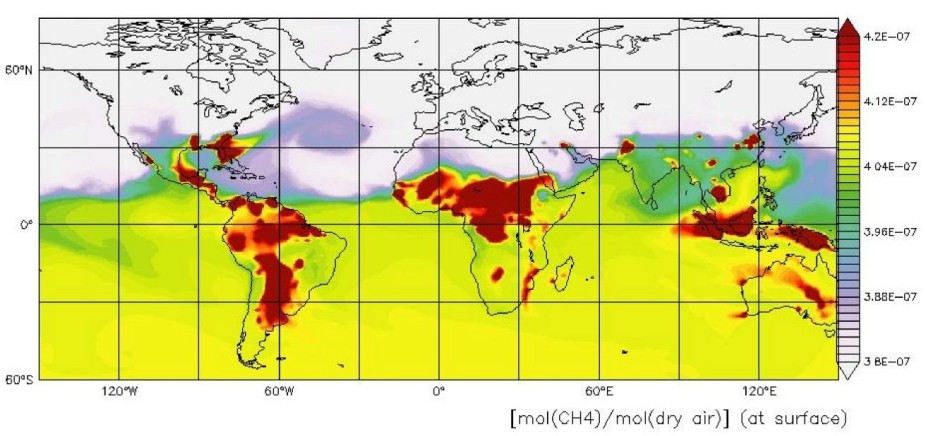


Figure 12:

a. Assumed additional emissions from tropical wetlands (TRO) as a possible explanation for rising methane after 2006, shown by the of the surface layer volume mixing-ratio.





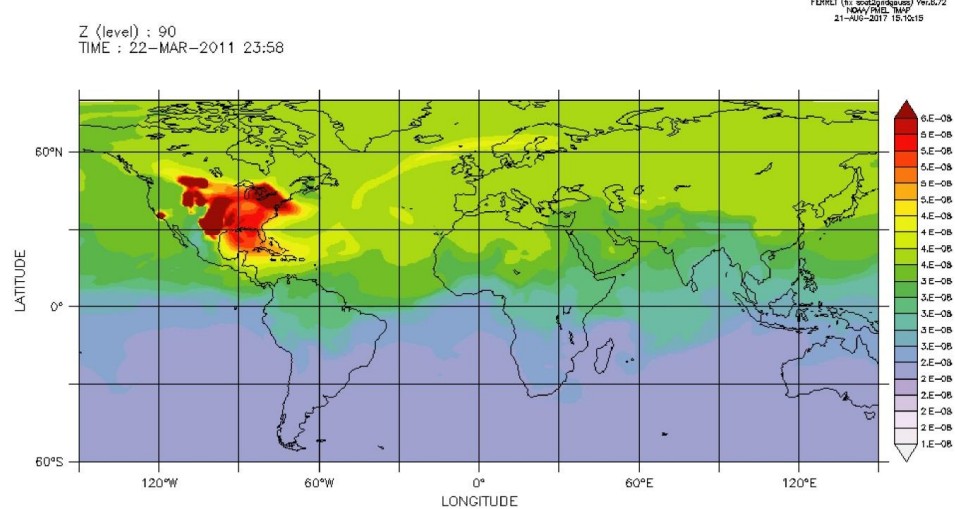

shale gas CH4 incr.near surface


b: Same as a, but for North American shale gas fracturing emissions (SHA).



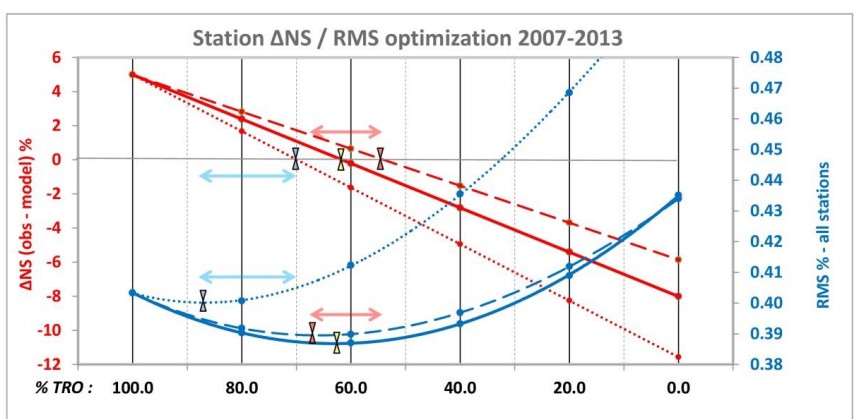

Figure 13: Influence of TRO/SHA ratio on the interhemispheric CH$_4$-gradient and all-station RMS:

785  a. Observation to simulation RMS (blue) and ΔNS (red) as a function of TRO to SHA ratio:

- solid lines : SHA scenario with emissions from the North American continent
- dashed lines : simulation results from SHA emissions over East Asia (FAE)
- dotted lines : simulation results from SHA emissions over Europe (EUR).





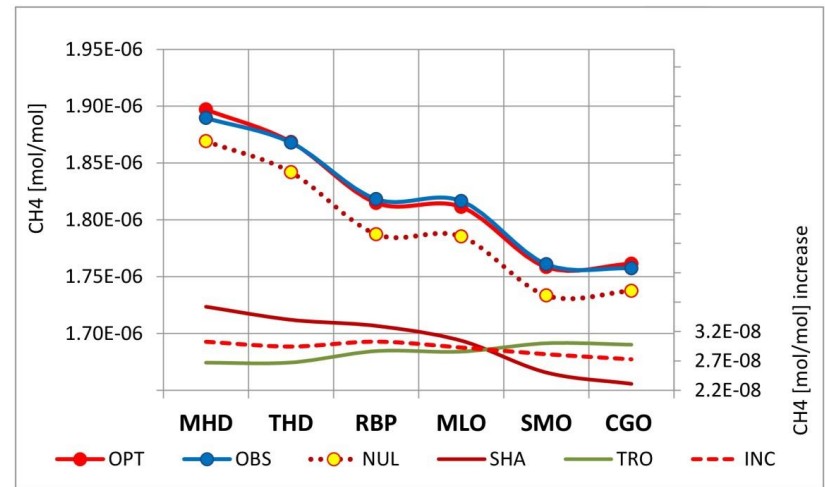


b.  Measured 2007 - 2013 average methane mixing ratios at the stations (blue dots) compared to the respective
    simulations without any emission increment (yellow dots) and with the optimal TRO/SHA-increment (red
    dots). The stations are ordered by latitude. The lower solid lines show the maximum possible contribution
    at the stations by the TRO (green) and SHA (red) emissions (i.e. assuming both emitting 28.3 Tg/yr), while

the red dashed depicts the increment using the 63/37 % fraction of these emissions, respectively.





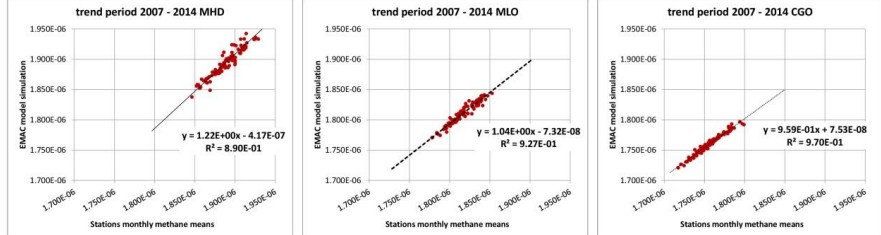

Figure 14: Regression analysis of EMAC calculations vs. observations of $CH_4$ at AGAGE/NOAA stations
MHD, MLO, and CGO for the trend years 2007 through 2014.




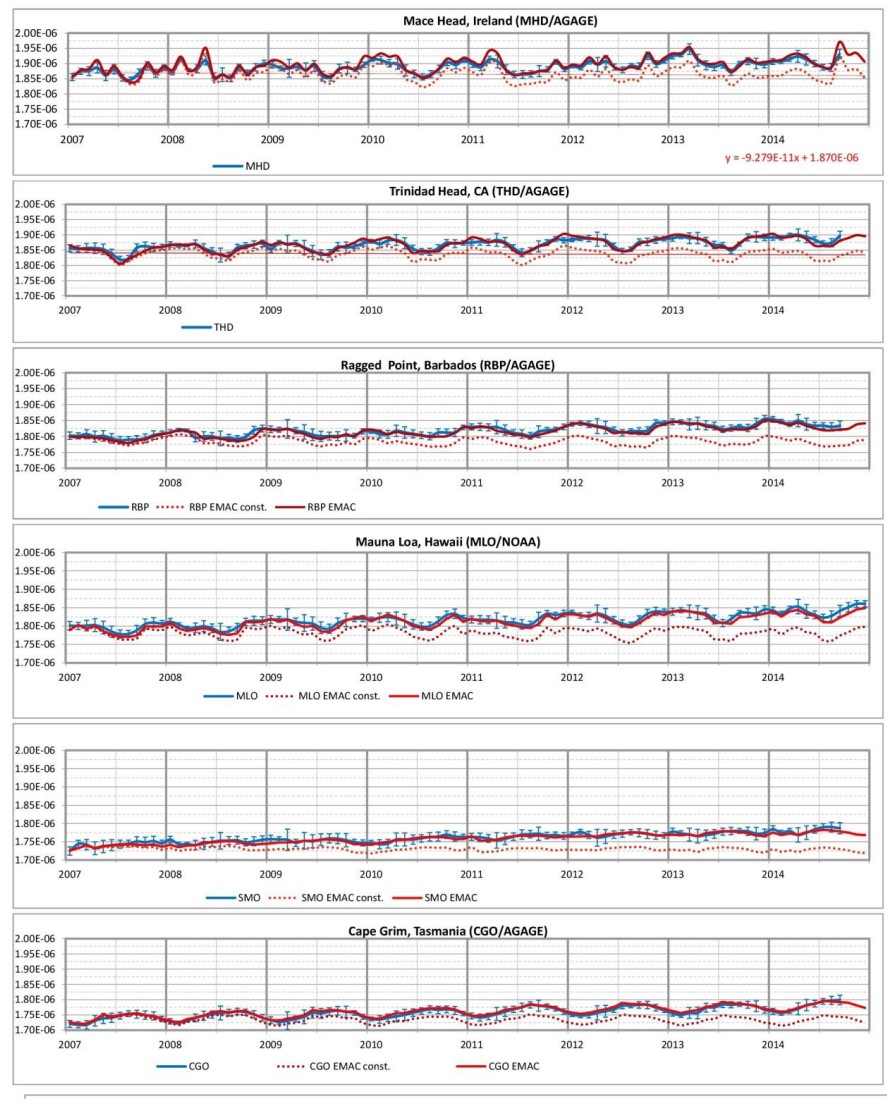

Figure 15:   CH$_4$ development at AGAGE/NOAA stations 2007 through 2014:

Observations (blue) vs. optimized TRO/SHA emission increment simulations (red, solid) and simulation without

any increment (red, dotted).





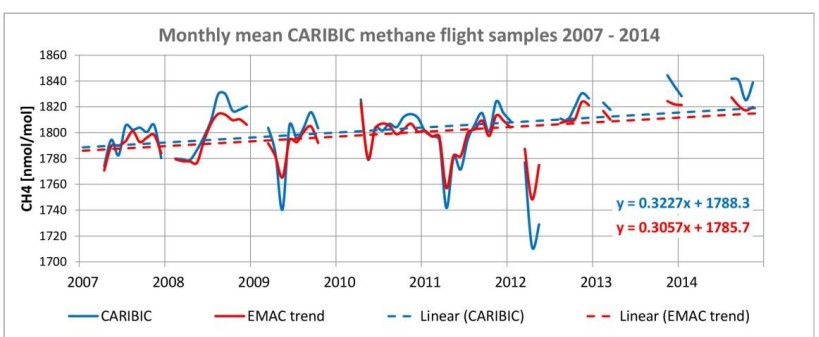


Figure 16: Monthly averaged EMAC-CH$_4$, including trend and CARIBIC-2 observations 2007 through 2014 for
all data obtained from CARIBIC whole air samples (WAS) in blue, and model results in red.





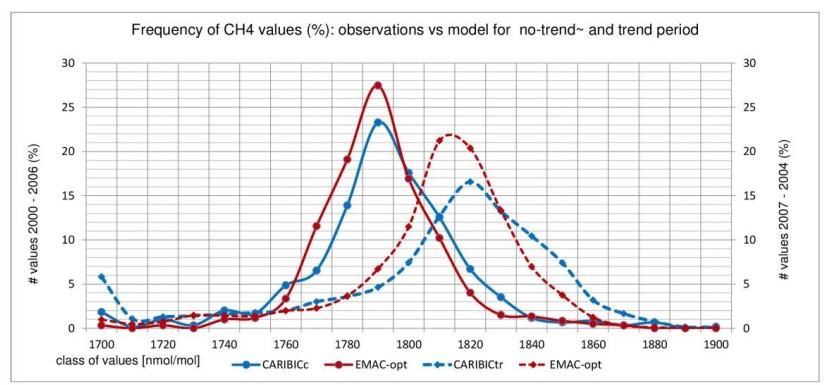

Figure 17: Frequency spectrum of CARIBIC observed and EMAC simulated $CH_4$-mixing-ratios separately

plotted for the years 2000-2006 and 2007-2014.




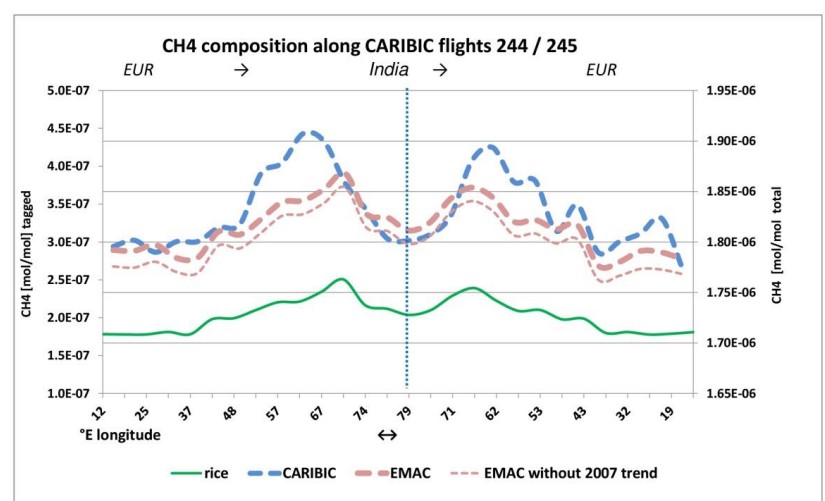

Figure 18:

a: CH$_4$ mixing ratios observed by CARIBIC (blue dashed, right axis) and calculated by EMAC (red dashed fat)
       and tagged rice related CH$_4$ (green, left axis) - India flights Aug. 2008. The thin red dashed line shows the
       simulation without trend period increment for reference.

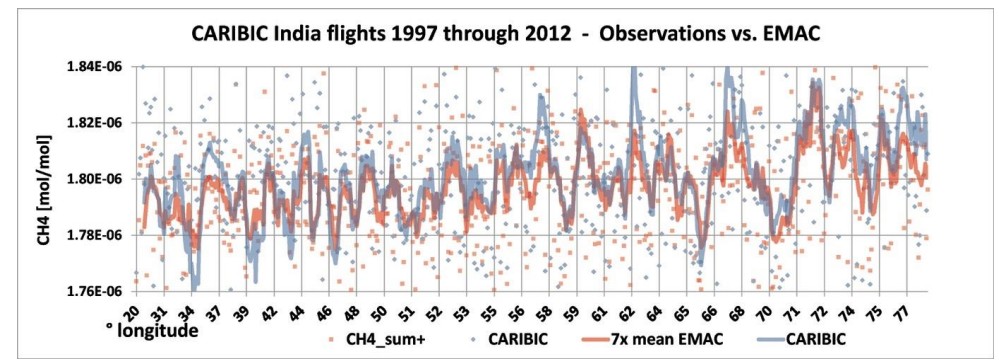

       b: CH$_4$ mixing ratios [mol/mol] observed by CARIBIC (blue) during all India flights 1997 through 2012 and
       corresponding EMAC simulations (red). The large scatter requires the sliding average of 7 points (solid
       lines).





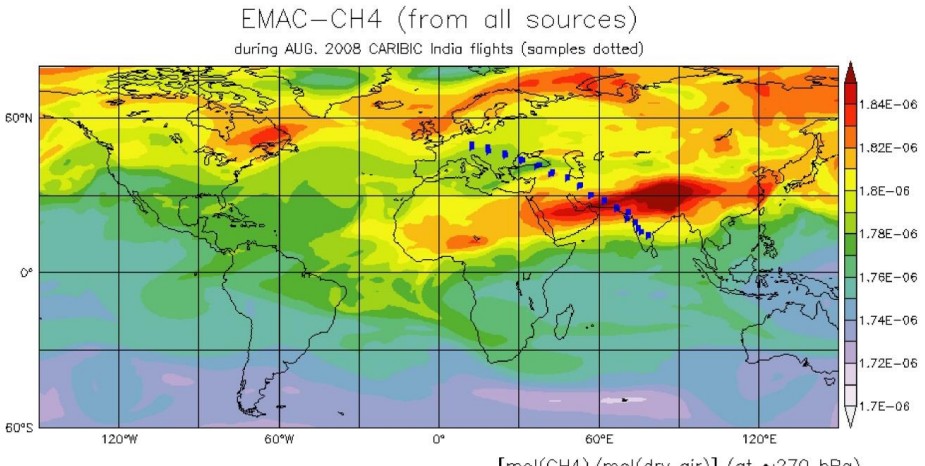


Figure 19:

a: Total EMAC-simulated CH$_4$ mixing ratios during the CARIBIC India flights 244/245 at cruise altitude.



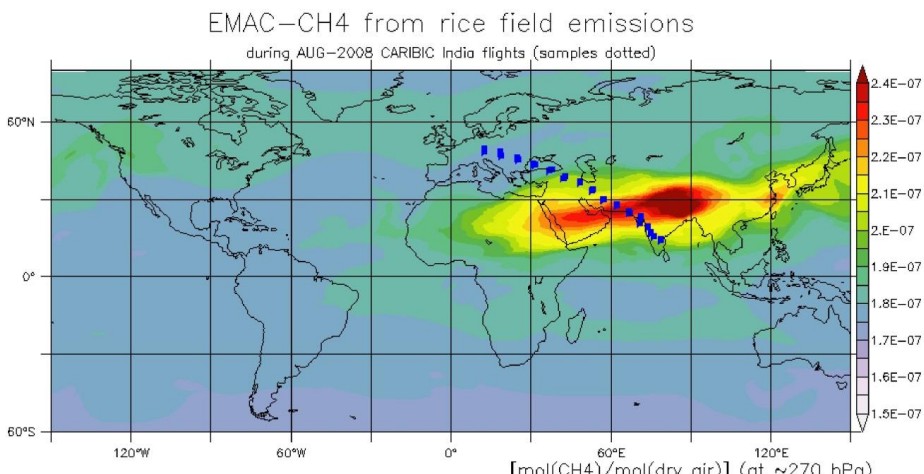

b: Tagged rice paddy released CH$_4$ in mixing ratios during CARIBIC India flights 244/245 at cruise altitude.





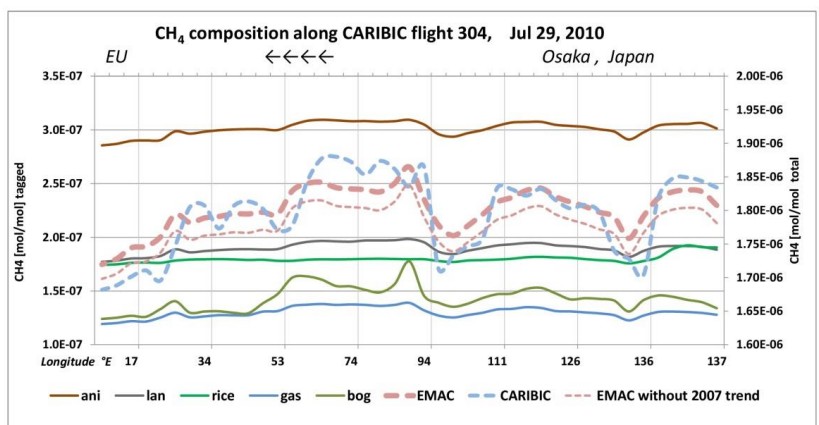

Figure 20: same as Fig. 18a but for Far East flight 304 July 2010. Tagged results are plotted for gas-, bog-, rice-,
835       landfill-, and animal (ruminant) released methane (left ordinate).