# Peer review of "Model simulations of atmospheric methane 1997-2016 and their evaluation using NOAA and AGAGE surface- and IAGOS-CARIBIC aircraft observations"

_Atmospheric Chemistry and Physics, 2017_

## Referee Comment (RC1) · Anonymous Referee #1 · 6 Apr 2018

General comments

The discussion paper of Zimmermann et al. presents an analysis of the global budget and trends of atmospheric $CH_4$ for the period 1997-2014, using the EMAC atmospheric chemistry general circulation model. As such the study contributes to the highly controversial discussion on the drivers of the renewed increase of atmospheric $CH_4$ observed since 2007, and is well within the scope of ACP. However, there are several significant limitations of the study, which limit the conclusions that can be drawn from the presented results.

(1) The study uses only a very limited number of atmospheric stations. In fact, only one single NOAA station (MLO) has been used (in addition to the 5 AGAGE stations). These 5+1 stations cover only the latitude range between 53$^o$N and 41$^o$S. It is not clear, why the authors do not use any data from the comprehensive NOAA ESRL global cooperative air sampling network (nor from the second NOAA station with continuous CH$_4$ measurements at Barrow, Alaska). The very limited set of stations used in this study limits the information that can be obtained on the CH$_4$ emissions at continental scale.

(2) 4 of the 6 stations used in this study are coastal sites (MHD, THD, RPB, CGO). Using such data requires that the model can properly simulate synoptic scale variability (e.g. change between marine and continental air masses). The EMAC model, however, is a general circulation model, and - as described in the paper - nudged to ECMWF meteorology only in the free troposphere (apart from surface pressure). Therefore, the capability of the EMAC model to simulate synoptic variability is probably worse compared to offline atmospheric transport models which are directly driven by analyzed meteorological fields. Good model representation of the continental stations, however, is essential for the study, since the interhemispheric gradient is derived as "as the difference between average CH$_4$ mixing-ratios at the northern stations MHD (53$^o$N) and THD (41$^o$N) and the southern station CGO (41$^o$S)" (lines 302-304) - and the inter-hemispheric gradient derived in this way is used to optimize the contribution from the "tropical wetland source (SWA)" and "landfill-, coal-, gas-, and oil (FOS)" emissions. Related to the concern of the potential limitations of the EMAC model to simulate the synoptic variability is the fact that the study uses "Monthly mean mixing - unfiltered with respect to local pollution events" (lines 205-206) measurements, which are compared to monthly mean model output. Especially for the 4 coastal sites, it would be more appropriate to use hourly (or daily) observations. If the EMAC model cannot properly simulate these sites, the use of monthly mean values for the comparison is likely to result in biased results.

(3) Unfortunately, the study investigates only 2 scenarios to analyze the recent $CH_4$ trend: (1) scenario "TRO" with additional emissions from the tropical wetlands, and (2) scenario "SHA" with additional emissions from the North American shale gas drilling sites. However, further hypotheses have been proposed in the literature, including increasing $CH_4$ emissions from agriculture and waste sectors [Saunois et al., 2017; Schaefer et al., 2016], and decreasing $CH_4$ emissions from biomass burning [Saunois et al., 2017; Worden et al., 2017]. While the decreasing $\delta^{13}CH_4$ observed in the atmosphere points to an increasing microbial sources (including both wetlands and anthropogenic microbial sources), Saunois et al. [2017] and Schaefer et al. [2016] concluded that among the microbial sources agriculture and waste sectors are more important than natural wetlands. This hypothesis is also supported by statistical data which suggest a significant increase of global CH4 emission from enteric fermentation and manure by 10 Tg $CH_4$ yr-1 between 2000 and 2011 ([Saunois et al., 2017], Fig. S12). The magnitude of the estimated decrease in biomass burning is smaller (estimated to be 3.7 ($\pm$1.4) Tg $CH_4$ per year from the 2001–2007 to the 2008–2014 period [Worden et al., 2017]), but plays an essential role for the $\delta^{13}CH_4$ budget and to reconcile the different hypotheses about the recent $CH_4$ increase.

Based on these general comments, I recommend to thoroughly revise the study, analyzing in more detail the capability of EMAC to simulate synoptic scale variability, to use a more comprehensive set of surface observations, and to include additional scenarios (in particular including the increase of $CH_4$ emissions from agricultural sources).

Further specific comments:

Abstract, line 21: I would suggest to replace "atmospheric CH4 calculations" by "atmospheric CH4 concentrations" or "atmospheric CH4 dry air mole fractions"

Abstract, line 24: "rescaling of individual emissions with proportional effects on the corresponding inventories": it is not clear what is meant here with "inventories" as compared to the "emissions".

Abstract, line 27: "all-station mean dry air mole fraction of 1792 nmol/mol": reference time period should be given (is this the 2000-2005 period mentioned in the following sentence, or the 1997-2006 period mentioned earlier?).

Abstract, line 38: "The coefficient of determination of R2 = 0.91 indicates even higher significance than before 2006": This could be partly due a larger range of concentrations values (and the given RMS is slightly higher than before 2006, indicating rather slightly poorer agreement).

Abstract, line 40-41: "...indicating that the model reproduces the seasonal and synoptic variability of CH4 in the upper troposphere and lower stratosphere." The analysis in the paper shows also clear limitations to simulate the variability in the lower stratosphere. This should be mentioned also in the abstract.

Introduction, lines 45-46: "and its concentration has been growing by about 1%/y since the beginning of the Anthropocene in the 19th century (Crutzen, 2002)": I would suggest to add further references for the atmospheric CH4 increase.

Introduction, lines 47-50: I would propose to present here mainly the most recent estimates of the radiative forcing. If the authors want to include also the older estimates, they should briefly explain the reasons for the large differences in the estimates. Furthermore, the given values "0.57 Wm−2 (direct 0.44Wm−2, indirect 0.13Wm−2)" are not consistent with the given [Dlugokencky et al., 2011] reference (where higher values are reported).

Introduction, lines 52-53: "...in 2007 the CH4 increase resumed unexpectedly (Bergamaschi et al., 2013)": Include here the primary references reporting the CH4 increase from the measurements ([Dlugokencky et al., 2009; Rigby et al., 2008]).

Introduction, lines 72-73: " Schaefer et al. (2016)... raising concern about the contribution from rice production versus wetland emissions". It should be mentioned here that Schaefer et al. (2016) conclude that the increase could be largely explained by

increase of CH4 emissions from ruminants (see also my general comment (3)): " Inventories report increased annual agricultural emissions over the 2000-2006 average of 12 Tg by 2011; dominated by ruminants (21, 23). This can largely account for the post-2006 [CH4]-growth, estimated at 15-22 Tg/a (30). Also, India and China's dominance in livestock-emissions (23) and S.E. Asian rice cultivation are consistent with the location of the source increase (13)."

Introduction, line 77: "Further, it was concluded that fossil fuel related sources had decreased". It should be stated explicitly who concluded this (it is not clear if this refers only to the [Schwietzke et al., 2016] or to both papers discussed here).

Model Setup, lines 117-118: "...operational analysis data of the European Centre for Medium-range Weather Forecasting (ECMWF) (van Aalst et al., 2004).": Why did the authors use operational analysis data and not the reanalysis (which should be superior in terms of consistency over time, which is essential for any trend analysis)?

Model Setup, lines 119-122: " the nudging method is applied in the free troposphere, tapering off towards the surface and tropopause, so that stratospheric dynamics are calculated freely, and possible inconsistencies between the boundary layer representations of the ECMWF and ECHAM models are avoided.": This might be an advantage in terms of self-consistency of the model physics, but may lead to deficiencies to simulate the synoptic-scale variability also in the boundary layer. As outlined in my general comment (2), the capability to simulate the synoptic-scale variations observed at the surface stations needs to be further analyzed (as this is essential to properly simulate the coastal stations used in this study).

Model Setup, line 127: "photolysis": Is this relevant in the EMAC model domain ?

Model Setup, lines 146-147: "Natural wetland emissions are based on Walter et al. (2000) and Fung et al. (1991).": These are different wetland inventories - which one has been used in this study ? Furthermore, the Walter et al. (2000) reference is missing.

Model Setup, lines 153: "GFED statistics": The specific GFED version number should be mentioned.

Model Setup, lines 154: "EDGAR2.0 database (Olivier, 2001)": Why has this old version of the EDGAR database been used, and not more recent versions?

Model Setup, lines 161: "yearly differences in the 20 Tg/y biomass burning": I would suggest to replace "yearly" by e.g. "inter-annual".

Model Setup, lines 179-180, "The negative flux distribution has a pronounced seasonal cycle in phase with the emissions": which emissions are meant here?

Observations used for model verification, line 190: Maybe replace "verification" by "validation" (however, there is indeed not a consistent use of these terms in the scientific literature)

Observations used for model verification, lines 199-205: The calibration scales used should be mentioned, including potential differences between the NOAA and AGAGE scales.

Observations used for model verification, lines 199-205: Why has only this very limited set of atmospheric stations been used (see general comment (1))?

Observations used for model verification, lines 205-207: "Monthly mean mixing - unfiltered with respect to local pollution events - are compared to respective monthly averaged model samples...": Why did the authors use monthly mean values, and not hourly or daily averages (see general comment (2)) ?

Observations used for model verification, lines 209ff: which calibration scale has been used for the CARIBIC CH4 measurements?

Simulation results, lines 229-230, "spin-up simulations and scaled to match the 1997 station measurements", and lines 254-255 " For initialization, a global methane distribution pattern for January was created iteratively in several spin-up cycles and finally

rescaled to Jan. 1997 station measurement data": The spin-up and scaling should be described in more detail (but best in section with model description): how long is the spin-up, which emissions have been used (probably the same as for the period 1997-2006?)? Did you just scale the calculated 3D fields? If so, there would be some inconsistency between the applied emissions and the concentrations (which may also explain why the simulated CH4 concentrations still increase between 1997 and 2000).

Simulation results, lines 265-269: "The linear dependency between source strength and atmospheric abundance...", and lines 286-289: "The integrated model CH4 masses exactly match the mass calculated": this has already been discussed before.

Simulation results, line 310: "... fossil group of categories comprising landfill-, coal-, gas-, and oil (FOS)": CH4 from landfills are (usually) not fossil, but primarily from relatively recent carbon.

Simulation results, line 346: "In contrast to the monthly average station data, the CARIBIC individual methane observations...": The station data - as provided to users - are hourly data. See also general comment (2).

Simulation results, lines 365-366: "...suggesting that the vertical resolution of the model grid is not optimal to resolve the fine structure in the tropopause region.": Probably this is not only due to coarse vertical resolution, but also due the vertical CH4 gradient in the stratosphere.

Simulation results, line 369: "according to the definition in Sect. 3.2 (Fig. S3)..." I assume this should be Fig. S2?

Simulation results, lines 386-387: "Figs. 12a and b...logarithmically scaled": the figures seem so use a linear scale. Furthermore, the figures show concentrations, while the figure caption states "Assumed additional emissions..." (Should be rephrased to e.g. "Impact of assumed additional emissions...").

Simulation results, lines 391-392: "...upper estimate from Bergamaschi et al. (2013) of

22 Tg CH4 yr-1 as a first guess": It should be mentioned that the estimate of Berga-maschi et al. (2013) is for a different time period (2007-2010), compared to the 2007-2014 period used in this study.

Simulation results, lines 395-396: "Both scenarios perfectly reproduce the observed CH4 trend...": I would suggest to avoid the term "perfectly".

Simulation results, lines 400-401: "Changes in the removal rate of methane by the OH radical have not been seen in other tracers of atmospheric chemistry, e.g. methyl chlo-roform (CH3CCl3) (Montzka et al., 2011; Lelieveld et al. 2016) and do not appear to explain short-term variations in methane.": I do not agree with this statement. Although Montzka et al. [2011] derive only small interannual variability their CH3CCl3 based es-timates still show variations on the order of +/- 3%, which is equivalent to a variability of the OH sink of +/- 17 Tg CH4 yr-1. Furthermore, the recent papers of Rigby et al. [2017] and Turner et al. [2017] demonstrated the potential significant impact of vari-ations in OH on the trend and interannual variability of CH4. I suggest to include the references to the two papers.

Simulation results, lines 423-424: "This shows that, when the SHA emissions are lo-cated away from the North America, no fraction is found that could minimize simulta-neously the $\Delta$NS and RMS": Given the very limited number of stations (see general comment (1)) and the question how well coastal / regional stations are simulated by the EMAC model (see general comment (2)), the question is, if this finding is really significant / robust.

Simulation results, line 437: "linear trend lines 0.32x (CARIBIC) and 0.31x (EMAC )": units are missing

Simulation results, lines 451-452: "the tropopause influence is stronger": probably also the influence of the lower stratosphere.

Conclusions and Outlook, lines 488-496: Would be useful to expand the conclusions,

including a discussion / summary of the novel aspects of this study, the uncertainties of the results and limitations of the study. Furthermore, it should be summarized, how the results from this study compare with the existing literature studies.

Conclusions and Outlook, lines 497-499: "In view of the additional global CH4 source since 2007, a source – sink equilibrium has not yet established after the 8 years of emissions considered. A 2nd order polynomial extrapolation predicts steady state after 13 years, assuming that the emissions remain unchanged.": This scenario seems quite hypothetical, and global emissions (including their latitudinal distribution) remaining constant over 13 years relatively unlikely.

Conclusions and Outlook, lines 497-499: "Nevertheless, the degree of freedom in the choice of sources is limited,...": Taking into account also uncertainties in the spatial (and temporal) distribution of emissions, a very large number of emission scenarios is possible - while only 2 scenarios were investigated in this study (see general comment (3)).

Table 1: references should be given for the individual a priori emission estimates.

Figures - general comment: The number of figures seems very large - several of them could be put in the supplementary material.

Figure 1: How were the data fitted?

Figure 3: Explain the meaning of the individual red circles

Figure 4a: "Zero" point (a priori emissions) should be indicated

Figure 4b: What is the meaning of the curves (interpolation) between the individual stations?

Figure 5: "dashed line": The figures seem not to show any dashed line.

Figure 7: What is the meaning of the colors?

Figure 11b: Is the shown average for all CARIBC flights as function of time really very useful? Probably the spatial coverage of the flights is also changing significantly over time.

Figure 13b: What is the meaning of the curves (interpolation) between the individual stations?

Figure 16: (as for Figure 11b): How is the spatial coverage of the flights changing over time?

Figure 17: Legend needs to be explained. Which curves are for which period?

References

Dlugokencky, E. J., L. Bruhwiler, J. W. C. White, L. K. Emmons, P. C. Novelli, S. A. Montzka, K. A. Masarie, P. M. Lang, A. M. Crotwell, J. B. Miller, and L. V. Gatti, Observational constraints on recent increases in the atmospheric CH4 burden, Geophys. Res. Lett., 36(L18803), doi: 10.1029/2009GL039780, 2009.

Dlugokencky, E. J., E. G. Nisbet, R. Fisher, and D. Lowry, Global atmospheric methane: budget, changes and dangers, Phil. Trans. R. Soc. A 369(1943), 2058-2072, doi: 10.1098/rsta.2010.0341, 2011.

Rigby, M., R. G. Prinn, P. J. Fraser, P. G. Simmonds, R. L. Langenfelds, J. Huang, D. M. Cunnold, L. P. Steele, P. B. Krummel, R. F. Weiss, S. O'Doherty, P. K. Salameh, H. J. Wang, C. M. Harth, J. Mühle, and L. W. Porter, Renewed growth of atmospheric methane, Geophys. Res. Lett., 35(L22805), doi:10.1029/2008GL036037, 2008.

Rigby, M., S. A. Montzka, R. G. Prinn, J. W. C. White, D. Young, S. O'Doherty, M. F. Lunt, A. L. Ganesan, A. J. Manning, P. G. Simmonds, P. K. Salameh, C. M. Harth, J. Mühle, R. F. Weiss, P. J. Fraser, L. P. Steele, P. B. Krummel, A. McCulloch, and S. Park, Role of atmospheric oxidation in recent methane growth, Proceedings of the National Academy of Sciences, 114(21), 5373-5377, doi: 10.1073/pnas.1616426114, 2017.

Saunois, M., P. Bousquet, B. Poulter, et al., Variability and quasi-decadal changes in the methane budget over the period 2000–2012, Atmos. Chem. Phys., 17(18), 11135-11161, doi: 10.5194/acp-17-11135-2017, 2017.

Schaefer, H., S. E. Mikaloff Fletcher, C. Veidt, K. R. Lassey, G. W. Brailsford, T. M. Bromley, E. J. Dlugokencky, S. E. Michel, J. B. Miller, I. Levin, D. C. Lowe, R. J. Martin, B. H. Vaughn, and J. W. C. White, A 21st century shift from fossil-fuel to biogenic methane emissions indicated by 13CH4, Science, doi: 10.1126/science.aad2705, 2016.

Turner, A. J., C. Frankenberg, P. O. Wennberg, and D. J. Jacob, Ambiguity in the causes for decadal trends in atmospheric methane and hydroxyl, Proceedings of the National Academy of Sciences, 114(21), 5367-5372, doi: 10.1073/pnas.1616020114, 2017.

Worden, J. R., A. A. Bloom, S. Pandey, Z. Jiang, H. M. Worden, T. W. Walker, S. Houweling, and T. Röckmann, Reduced biomass burning emissions reconcile conflicting estimates of the post-2006 atmospheric methane budget, Nature Communications, 8(1), 2227, doi: 10.1038/s41467-017-02246-0, 2017.
* * *

---

## Referee Comment (RC2) · Anonymous Referee #2 · 17 Apr 2018

In this manuscript the authors describe the global atmospheric $CH_4$ budget using the EMAC modelling system trying to understand and simulate the observed trends of the years 1997–2014 from a number of ground stations and a vast collection of aircraft observations.

The manuscript is rather well written and the results are well presented with a large amount of figures to support the text. Also there are interesting findings on the sources of $CH_4$ contributing to the existing knowledge around it. However there are a few points that need to be addressed before publication to ACP.

[Figure]

General Comments

The authors should really explain how the selection of stations was made. It is even stated in the title that they use NOAA stations for methane, but only the Mauna Loa Observatory is used. Also the number of stations seems quite limited to accurately represent the global methane.

The choice of meteorological data seems a bit strange. The operational data of ECMWF have changed vertical resolution at least twice within the study period, definitely affecting the height of each level. This must have an impact on the nudged values and the model results. How did you deal with these issues? Did the meteorological data vertical resolution near tropopause match the model vertical resolution? Also a validation of the computed meteorology is missing from the manuscript.

Specific Comments

P1 L25: RMS abbreviation used before defining.

P4 L153: Which GFED? GFED4s? Clearly state the version.

P5 L178: emission flux, the "e" is missing.

P7 L231: A higher resolution of sampling should be used for the CARIBIC data. Daily samples for flight data is far too long.

P8 L297-305: As mentioned in the general comments, 6 stations are not enough to reach definite conclusions.

P10 L360-368: Couldn't this be because of the meteo data?

P11 L418 and P12 L434 and P13 L495 and Fig13b caption: Be consistent when reporting these numbers.

Fig1: A different color code for the different periods would be helpful.

Fig4a and b: I believe the lines connecting the circles are misleading.

Figs 5, 6, 9, 11, 14, 15: Really hard to read because of size.

Fig13b: again I fail to see the need for the line connecting stations.

Fig17: State either in the caption or in the legend which set of lines is for every period.

Supplementary material:

There are some inconsistencies between the figures and the captions making it sometimes confusing.

---

## Author Comment (AC1) · 15 Jan 2019

We thank the referee for the comments.
Here the comments are listed with our reply.

**General Comments**

**The authors should really explain how the selection of stations was made. It is even stated in the title that they use NOAA stations for methane, but only the Mauna Loa Observatory is used. Also the number of stations seems quite limited to accurately represent the global methane.**

Indeed only a limited amount of stations were used in the comparison. Following the referee's advice, the simulation was repeated using 16 NOAA stations and CGO(AGAGE) :

| Code | Station Name | Country | Lat ° | Lon ° | elevation / m |
|------|-------------|---------|-------|-------|---------------|
| ALT | Alert | Canada | 82.45 | -62.51 | 190 |
| ASC | Ascension Island | UK | -7.97 | -14.40 | 85 |
| AZR | Terceira Ile., Azores | Portugal | 38.77 | -27.38 | 19 |
| BRW | Barrow, Alaska | USA | 71.32 | -156.61 | 11 |
| CGO | Cape Grim, Tasmania | Australia | -40.68 | 144.69 | 94 |
| CRZ | Crozet Island | France | -46.43 | 51.85 | 197 |
| EIC | Easter Island | Chile | -27.16 | -109.43 | 47 |
| GMI | Mariana Islands | Guam | 13.39 | 144.66 | 0 |
| HBA | Halley Station, | Antarctica, UK | -75.61 | -26.21 | 30 |
| MLO | Mauna Loa, Hawaii | USA | 19.54 | -155.58 | 3397 |
| RPB | Ragged Point | Barbados | 13.17 | -59.43 | 15 |
| SEY | Mahe Island, | Seychelles | -4.68 | 55.53 | 2 |
| SHM | Shemya Island, Alaska | USA | 52.71 | 174.13 | 23 |
| SMO | Tutuila, Am. Samoa | USA | -14.25 | -170.56 | 42 |
| SPO | South Pole | USA | -89.98 | -24.80 | 2810 |
| ZEP | Ny-Alesund, Svalbard | Norway, Sweden | 78.91 | 11.89 | 474 |

**The choice of meteorological data seems a bit strange. The operational data of ECMWF have changed vertical resolution at least twice within the study period, definitely affecting the height of each level. This must have an impact on the nudged values and the model results. How did you deal with these issues? Did the meteorological data vertical resolution near tropopause match the model vertical resolution? Also a validation of the computed meteorology is missing from the manuscript.**

We thank the referee for pointing this out. The new simulations has been performed using the ERA interim data (Dee et al., 2011), which is consistent for the entire simulation period.

**Specific Comments**

**P1 L25: RMS abbreviation used before defining.**
Abstract L27: Root Mean Square deviation (RMS)

**P4 L153: Which GFED? GFED4s? Clearly state the version.**
GFEDv4: Randerson, J.T., G.R. van der Werf, L. Giglio, G.J. Collatz, and P.S. Kasibhatla. 2018. Global Fire Emissions Database, Version 4, (GFEDv4). ORNL DAAC, Oak Ridge, Tennessee, USA.
https://doi.org/10.3334/ORNLDAAC/1293

**P5 L178: emission flux, the "e" is missing**. – this has been corrected in the manuscript.

**P7 L231: A higher resolution of sampling should be used for the CARIBIC data. Daily samples for flight data is far too long.**

We apologize for the unclear formulation: the model was sampled daily at 12 UTC at the Stations' location, while for comparison with CARIBIC data  the highest possible sampling was used (2 min time-step). Nevertheless, in the new simulation also station values are sampled continuously and then averaged monthly for comparison with observations.

"… calculated $CH_4$ mixing ratios are recorded and stored at all sampling positions and -times at selected (NOAA (Dlugokencky, 2018) and AGAGE (Prinn et al., 2013) observation sites and along the CARIBIC flight tracks"

**P8 L297-305: As mentioned in the general comments, 6 stations are not enough to reach definite conclusions.**

This has been changed in the largely revised manuscript.

**P10 L360-368: Couldn't this be because of the meteo data?**

We exchanged the "operational analyses" with ERA interim with the same effect.
However the high altitude mixing-ratios are averaged over ~500 m grid boxes smoothing down the amplitudes.

**P11 L418 and P12 L434 and P13 L495 and Fig13b caption: Be consistent when reporting these numbers.**

We corrected the caption.

"…For the years 2007 through 2013 it turns out that a total emission of 25.47 Tg $CH_4$/y composed of 19.44 Tg TRO and 5.74 Tg SHA optimally reduces the RMS to 0.55 % and approximates the observed ΔNH/SH up to 98%. Fig, 10 …"

[Figure]

Figure 10: Scaling TRO and SHA emission fractions to fit the all-station observations within smallest RMS:
Left: Observations (blue) and  total calculated  $CH_4$  without- (black), and with (red) trend period emissions (solid lines right panel).

Right: A-priori estimates (dashed)  and  solver-scaled (solid) TRO (19.44)- and SHA (5.74 Tg/y)
emissions for trend years.

**Fig1: A different color code for the different periods would be helpful.**

Following referee's suggestion, we change the figure.

[Figure]

**Fig4a and b: I believe the lines connecting the circles are misleading.**

New simulation: (I am not sure if looks is better . . . ?)

[Figure]

**Figs 5, 6, 9, 11, 14, 15: Really hard to read because of size.**

Vector graphics of all figures will be provided for the final publication.

**Fig13b: again I fail to see the need for the line connecting stations.**

New simulation: (I am not sure if looks is better . . . ?)

[Figure]

**Fig17: State either in the caption or in the legend which set of lines is for every period.**

Based on the results of the new simulation, the figure (and caption) has been updated.

[Figure]

Figure 15: Frequency spectrum of CARIBIC observed and EMAC simulated CH4-mixing-ratios separately plotted for the years 2000-2006 and 2007-2014.

**Supplementary material:**

**There are some inconsistencies between the figures and the captions making it sometimes confusing.**

The supplement has been updated.,

---

## Author Comment (AC2) · 31 Jan 2019

We thank the referee for the comments.
Here the comments are listed (black) with our reply (red, italics).

General comments
The discussion paper of Zimmermann et al. presents an analysis of the global budget and trends of atmospheric CH4 for the period 1997-2014, using the EMAC atmospheric chemistry general circulation model. As such the study contributes to the highly controversial discussion on the drivers of the renewed increase of atmospheric CH4 observed since 2007, and is well within the scope of ACP. However, there are several significant limitations of the study, which limit the conclusions that can be drawn from the presented results.
*Following the referee's advice, most points of criticism have been taken into account and new simulation runs have been performed:*

(1) The study uses only a very limited number of atmospheric stations.
*Indeed only a limited amount of stations were used in the comparison. Following the referee's advice, the simulation was repeated using 16 NOAA stations and CGO (AGAGE):*

| Code | Station Name | Country | Lat ° | Lon ° | elevation / m |
|------|--------------|---------|-------|-------|---------------|
| ALT | Alert | Canada | 82.45 | -62.51 | 190 |
| ASC | Ascension Island | UK | -7.97 | -14.40 | 85 |
| AZR | Terceira Ile., Azores | Portugal | 38.77 | -27.38 | 19 |
| BRW | Barrow, Alaska | USA | 71.32 | -156.61 | 11 |
| CGO | Cape Grim, Tasmania | Australia | -40.68 | 144.69 | 94 |
| CRZ | Crozet Island | France | -46.43 | 51.85 | 197 |
| EIC | Easter Island | Chile | -27.16 | -109.43 | 47 |
| GMI | Mariana Islands | Guam | 13.39 | 144.66 | 0 |
| HBA | Halley Station, | Antarctica, UK | -75.61 | -26.21 | 30 |
| MLO | Mauna Loa, Hawaii | USA | 19.54 | -155.58 | 3397 |
| RPB | Ragged Point | Barbados | 13.17 | -59.43 | 15 |
| SEY | Mahe Island, | Seychelles | -4.68 | 55.53 | 2 |
| SHM | Shemya Island, Alaska | USA | 52.71 | 174.13 | 23 |
| SMO | Tutuila, Am. Samoa | USA | -14.25 | -170.56 | 42 |
| SPO | South Pole | USA | -89.98 | -24.80 | 2810 |
| ZEP | Ny-Alesund, Svalbard | Norway, Sweden | 78.91 | 11.89 | 474 |

In fact, only one single NOAA station (MLO) has been used (in addition to the 5 AGAGE stations). These 5+1 stations cover only the latitude range between 53oN and 41oS. It is not clear, why the authors do not use any data from the comprehensive NOAA ESRL global cooperative air sampling network (nor from the second NOAA station with continuous CH4 measurements at Barrow, Alaska). The very limited set of stations used in this study limits the information that can be obtained on the CH4 emissions at continental scale.

(2) 4 of the 6 stations used in this study are coastal sites (MHD, THD, RPB, CGO). Using such data requires that the model can properly simulate synoptic scale variability (e.g. change between marine and continental air masses).

*We thank the referee for pointing this out.*
*In the new simulation all stations are "clean air" sites and filtered wrt synoptic scale pollution (except CGO).*
*We also revised the text in Ch. 3.1:*
*"The data provided (Dlugokencky et al., 2018) are filtered with respect to synoptic scale pollution events. We take advantage of 16 stations about fairly equally distributed over the globe (Fig. 2a) and remote from the major emission areas to ensure comparability with the model results which are not filtered. For the same reason, in case of Cape Grim, Australia (41° S, 145°) we refer to the unfiltered AGAGE records (Prinn et al., 1978, 2013). At all stations monthly mean mixing-ratios are compared to respective monthly averaged model samples. "*

*Our actual model resolution "T106" with a grid size of 125 km at the Equator is able to capture also synoptic scale events; however the fifteen NOAA stations that we used are remote from the main source regions. CGO is influenced my air from the Australian continent but the model is able to reproduce eventual pollution events.*

The EMAC model, however, is a general circulation model, and - as described in the paper – nudged to ECMWF meteorology only in the free troposphere (apart from surface pressure). Therefore, the capability of the EMAC model to simulate synoptic variability is probably worse compared to offline atmospheric transport models which are directly driven by analyzed meteorological fields.

*Indeed, we operate the EMAC GCM/CCM in "nudged" mode, i.e. by Newtonian relaxation towards ECMWF meteorology. The nudging (of divergence, vorticity, temperature (excluding global mean) and logarithm of surface pressure) is applied in spectral space, however (by so-called low normal mode insertion) only down to the synoptic scale. Thus, the meteorological sequence of ECMWF is reproduced by EMAC on the synoptic scale, whereas the variability on sub-synoptic scale is determined by the model physics (e.g. convection etc.).*
*Thus, the synoptic variability should not be an issue here.*

Good model representation of the continental stations, however, is essential for the study, since the is derived as "as the difference between average CH4 mixing-ratios at the northern stations MHD (53oN) and THD (41oN) and the southern station CGO (41oS)" (lines 302-304) - and the interhemispheric gradient derived in this way is used to optimize the contribution from the "tropical wetland source (SWA)" and "landfill-, coal-, gas-, and oil (FOS)" emissions.

*Following the referee's advice, in the new simulation we are using an improved interhemispheric gradient definition and define:*
*ΔNH/SH = avg(ALT, ZEP, BRW) minus avg(CGO, CRZ, HBA, SPO).*

*Furthermore the contributions of all (10) emission categories are now explicitly considered in the optimization procedure.*

Related to the concern of the potential limitations of the EMAC model to simulate the synoptic variability is the fact that the study uses "Monthly mean mixing - unfiltered with respect to local pollution events" (lines 205-206) measurements, which are compared to monthly mean model output. Especially for the 4 coastal sites, it would be more appropriate to use hourly (or daily) observations. If the EMAC model cannot properly simulate these sites, the use of monthly mean values for the comparison is likely to result in biased results.

*As mentioned above, the sites have changed in the new simulation.*
*Consistently with the measurements, the model now samples at every time step and averages at the end of every month (sub-model SCOUT).*

 (3) Unfortunately, the study investigates only 2 scenarios to analyze the recent CH4 trend: (1) scenario "TRO" with additional emissions from the tropical wetlands, and (2) scenario "SHA" with additional emissions from the North American shale gas drilling sites. However, further hypotheses have been proposed in the literature, including increasing CH4 emissions from agriculture and waste sectors [Saunois et al., 2017; Schaefer et al., 2016], and decreasing CH4 emissions from biomass burning [Saunois et al., 2017; Worden et al., 2017].

*We do not discriminate the latter sectors in this study. It has been our intension to test the sensitivity of the station- and flight records to the hypothetical TRO and SHA emission assumptions.*
*Biomass burning emissions are not considered in the trend phase.*
*We take advantage of GFED4.1s which includes agricultural waste burning. No persistent decrease is obvious in this dataset (Fig. 1 at the end of this document.)*

While the decreasing _13CH4 observed in the atmosphere points to an increasing microbial sources (including both wetlands and anthropogenic microbial sources), Saunois et al. [2017] and Schaefer et al. [2016] concluded that among the microbial sources agriculture and waste sectors are more important than natural wetlands.

*It is not possible to resolve the latter sectors in the current model setup. The referee's advice will be subject to further investigation also considering observations beyond 2016.*

This hypothesis is also supported by statistical data which suggest a significant increase of global CH4 emission from enteric fermentation and manure by 10 Tg CH4 yr-1 between 2000 and 2011 ([Saunois et al., 2017], Fig. S12). The magnitude of the estimated decrease in biomass burning is smaller (estimated to be 3.7 (±1.4) Tg CH4 per year from the 2001–2007 to the 2008–2014 period [Worden et al., 2017]), but plays an essential role for the _13CH4 budget and to reconcile the different hypotheses about the recent CH4 increase.

Based on these general comments, I recommend to thoroughly revise the study, analyzing in more detail the capability of EMAC to simulate synoptic scale variability, to use a more comprehensive set of surface observations,

*Following the referee's advice we revised the study and performed new simulations.*
and to include additional scenarios (in particular including the increase of CH4 emissions from agricultural sources).

*It has been our aim from the beginning to test how well the taken assumptions (in form of given emission datasets of 10 categories) can explain the observations. Realizing a significant underestimation from 2007 on, the two hypothetical scenarios TRO and SHA have been included for sensitivity testing of the observed trend.*

Further specific comments:

Abstract, line 21: I would suggest to replace "atmospheric CH4 calculations" by "atmospheric CH4 concentrations" or "atmospheric CH4 dry air mole fractions"

*Ok: "The atmospheric $CH_4$ dry air mole fractions . . ."*

Abstract, line 24: "rescaling of individual emissions with proportional effects on the corresponding inventories": it is not clear what is meant here with "inventories" as compared to the "emissions".

*New formulation: "... rescaling of individual emissions with proportional effects on respective source segregated methane abundances."*

Abstract, line 27: "all-station mean dry air mole fraction of 1792 nmol/mol": reference time period should be given (is this the 2000-2005 period mentioned in the following sentence, or the 1997-2006 period mentioned earlier?).

*New formulation: ". . . the 2000 – 2005 observed all-station mean dry air mole fraction of 1780 nmol/mol could be reproduced within an RMS = 0.40 % . . ."*

Abstract, line 38: "The coefficient of determination of R2 = 0.91 indicates even higher significance than before 2006": This could be partly due a larger range of concentrations values (and the given RMS is slightly higher than before 2006, indicating rather slightly poorer agreement).

*In the new simulation including 16 ground stations we had to update the statistics:*
*"We explored the contributions of two potential causes, one representing natural emissions from wetlands in the tropics "TRO", and the other anthropogenic (e.g. shale gas fracturing) emissions in North America "SHA". Based on the acceptance of the no-trend period emission distribution, with the Solver we estimated annual 19.4 TRO and 5.7 Tg/y SHA contributions, respectively, to optimally fit the trend (RMS = 0.55 % / $R^2$ = 0.88). "*

Abstract, line 40-41: "...indicating that the model reproduces the seasonal and synoptic variability of CH4 in the upper troposphere and lower stratosphere." The analysis in the paper shows also clear limitations to simulate the variability in the lower stratosphere. This should be mentioned also in the abstract.

*Following the referee's advice we included the following statement:*
*"The coefficient of determination $R^2$ implies that the model reproduces the seasonal and synoptic variability of $CH_4$ in the UTLS. Regression analysis however indicates evident underestimation of the calculated $CH_4$ variability, suggesting that the vertical resolution of the model grid is not optimal to resolve the fine structure in the tropopause region. "*

Introduction, lines 45-46: "and its concentration has been growing by about 1%/y since the beginning of the Anthropocene in the 19th century (Crutzen, 2002)": I would suggest to add further references for the atmospheric CH4 increase.

*Following the referee's advice we included: "Clais et al. 2013"*

*Reference: Ciais, P., C. Sabine, G. Bala, L. Bopp, V. Brovkin, J. Canadell, A. Chhabra, R. DeFries, J. Galloway, M. Heimann, C. Jones, C. Le Quéré, R.B. Myneni, S. Piao and P. Thornton, 2013: Carbon and Other Biogeochemical Cycles. In: Climate Change 2013: The Physical Science Basis. Contribution of Working Group I to the Fifth Assessment Report of the Intergovernmental Panel on Climate Change [Stocker, T.F., D. Qin, G.-K. Plattner, M. Tignor, S.K. Allen, J. Boschung, A. Nauels, Y. Xia, V. Bex and P.M. Midgley (eds.)]. Cambridge University Press, Cambridge, United Kingdom and New York, NY, USA.*

Introduction, lines 47-50: I would propose to present here mainly the most recent estimates of the radiative forcing. If the authors want to include also the older estimates, they should briefly explain the reasons for the large differences in the estimates. Furthermore, the given values "0.57 Wm−2 (direct 0.44Wm−2, indirect 0.13Wm−2)" are not consistent with the given [Dlugokencky et al., 2011] reference (where higher values are reported).

*Following the referee's advice we included:*

*"The resulting factor of 2.5 increase in the global abundance of atmospheric methane ($CH_4$) since 1750 contributes 0.5 $Wm^{-2}$ to total direct radiative forcing by long-lived greenhouse gases (2.77 $Wm^{-2}$ in 2009), while its role in atmospheric chemistry adds another approximately 0.2 $Wm^{-2}$ of indirect forcing (Dlugokencky et al., 2011) Etminan et al, (2016) presented new calculations including the impact of the shortwave forcing and found that the 1750‑2011 radiative forcing is about 25% higher (increasing from 0.48Wm−2 to 0.61Wm−2) compared to the value in the Intergovernmental Panel on Climate Change (IPCC) 2013 assessment."*

*Reference: Etminan, M., G. Myhre, E. J. Highwood, and K. P. Shine (2016), Radiative forcing of carbon dioxide, methane, and nitrous oxide: A significant revision of the methane radiative forcing, Geophys. Res. Lett., 43, 12,614 – 12,623, doi:10.1002/2016GL071930.*

Introduction, lines 52-53: "...in 2007 the CH4 increase resumed unexpectedly (Bergamaschi et al., 2013)": Include here the primary references reporting the CH4 increase from the measurements ([Dlugokencky et al., 2009; Rigby et al., 2008]).

*Following the referee's advice we included these references:*

*"The resuming upward trend after 2007 (Dlugokencky et al., 2009; Rigby et al., 2008, IPCC 2014) is not fully understood: data analysis (Nisbet et al., 2016, Worden et al., 2017) and inverse modelling studies (Bergamaschi et al., 2013) indicate…"*

Introduction, lines 72-73: "Schaefer et al. (2016)... raising concern about the contribution from rice production versus wetland emissions". It should be mentioned here that Schaefer et al. (2016) conclude that the increase could be largely explained by increase of CH4 emissions from ruminants (see also my general comment (3)): " Inventories report increased annual agricultural emissions over the 2000-2006 average of 12 Tg by 2011; dominated by ruminants (21, 23). This can largely account for the post-2006 [CH4]-growth, estimated at 15-22 Tg/a (30). Also, India and China's dominance in livestock-emissions (23) and S.E. Asian rice cultivation are consistent with the location of the source increase (13)."

*We inserted in the paragraph before:*
*"Schaefer at al. (2016) by means of 13C/ 12C (CH4) data and a box model concluded that fossil fuel related emissions are a minor contributor to the renewed methane increase, compared to agricultural emissions dominated by ruminants."*

Introduction, line 77: "Further, it was concluded that fossil fuel related sources had decreased". It should be stated explicitly who concluded this (it is not clear if this refers only to the [Schwietzke et al., 2016] or to both papers discussed here).
*We updated this paragraph:*
*"As mentioned above, Schaefer et al. (2016) showed that "after 2006, the activation of biogenic emissions caused the renewed $CH_4$ rise", raising concern about the contribution from rice production versus wetland emissions, and Schwietzke et al. (2016), based on reassessment of data of the $^{13}C/^{12}C$ ratio of $CH_4$ from fossil sources, conclude that the assumed global fossil fuel $CH_4$ emissions need a major upward revision of 60-110 %. In other words, it was found by both authors that the combined fossil $CH_4$ sources (1985-2002) must have been much stronger (factor of 2), at the expense of microbial sources. Further, it was concluded that fossil fuel related sources had decreased."*

Model Setup, lines 117-118: "...operational analysis data of the European Centre for Medium-range Weather Forecasting (ECMWF) (van Aalst et al., 2004).": Why did the authors use operational analysis data and not the reanalysis (which should be superior in terms of consistency over time, which is essential for any trend analysis)?
*Following the referee's advice, we use the ERA-Interim coefficients in the new simulation.*
*Reference:*
*Dee, D. P., et al.: The ERA-Interim reanalysis: configuration and performance of the data assimilation system, Q. J. Roy. Meteor. Soc., 137, 553–597, doi:10.1002/qj.828, 2011.*

Model Setup, lines 119-122: " the nudging method is applied in the free troposphere, tapering off towards the surface and tropopause, so that stratospheric dynamics are calculated freely, and possible inconsistencies between the boundary layer representations of the ECMWF and ECHAM models are avoided.": This might be an advantage in terms of self-consistency of the model physics, but may lead to deficiencies to simulate the synoptic-scale variability also in the boundary layer. As outlined in my general comment (2), the capability to simulate the synoptic-scale variations observed at the surface stations needs to be further analyzed (as this is essential to properly simulate the coastal stations used in this study).
*see above  -  and no coastal stations anymore except CGO which is simulated at $R^2 = .93$*
*(Fig. 1 at the end of this document)*

Model Setup, line 127: "photolysis": Is this relevant in the EMAC model domain?
*Because the model domain reaches up to 1 Pa we mentioned that for completeness, even the effect is small.  "Removal of CH4 by photolysis becomes important only in the mesosphere."*
*(T. Röckmann et al., The isotopic composition of methane in the stratosphere:*

*high-altitude balloon sample measurements, Atmos. Chem. Phys., 11, 13287–13304, 2011 doi:10.5194/acp-11-13287-2011)*

Model Setup, lines 146-147: "Natural wetland emissions are based on Walter et al. (2000) and Fung et al. (1991).": These are different wetland inventories - which one has been used in this study ? Furthermore, the Walter et al. (2000) reference is missing.

*We updated this paragraph:*
*"Natural wetland emissions are based on Walter et al. (2000), fossil sources based on  EDGARV2.0 and remaining sources as compiled by I. Fung et al. (1991)."*

Model Setup, lines 153: "GFED statistics": The specific GFED version number should be mentioned.

*The new simulation is based on GFEDv4s.*
*Reference: Randerson, J. T., G.R. van der Werf, L. Giglio, G.J. Collatz, and P.S. Kasibhatla. 2018. Global Fire Emissions Database, Version 4, (GFEDv4). ORNL DAAC, Oak Ridge, Tennessee, USA. https://doi.org/10.3334/ORNLDAAC/1293*

Model Setup, lines 154: "EDGAR2.0 database (Olivier, 2001)": Why has this old version of the EDGAR database been used, and not more recent versions?

*We did not update this dataset because this category contributes just 2.6 % of total CH4 and because the dataset serves as a priory assumption for the optimization procedure.*

Model Setup, lines 161: "yearly differences in the 20 Tg/y biomass burning": I would suggest to replace "yearly" by e.g. "inter-annual".

*We follow the referee's suggestion.*

Model Setup, lines 179-180, "The negative flux distribution has a pronounced seasonal cycle in phase with the emissions": which emissions are meant here?

*In the new simulation the "negative flux" approach is replaced by a "deposition velocity" parametrization which was not yet ready for the old paper version.*
*We updated the respective paragraph:*
*"The MESSy sub-model "DDEP" simulates dry deposition of gas phase tracers and aerosols (Kerkweg et al. 2006). For our CH$_4$ budget modeling the deposition velocity was derived for a fixed atmospheric-methane mixing ratio of 1800 nmole/mole (Spahni R. et al., 2011, Ridgwell et al., 1999) and is scaled correspondingly. The deposition has a pronounced seasonal cycle in phase with the wetland emissions and depends on soil temperature, moisture content and the land cultivation fraction and varies from 2.4 Tg in January to 4.0 Tg in July."*

Observations used for model verification, line 190: Maybe replace "verification" by "validation" (however, there is indeed not a consistent use of these terms in the scientific literature)
*We follow the referee's suggestion.*

Observations used for model verification, lines 199-205: The calibration scales used should be mentioned, including potential differences between the NOAA and AGAGE scales.

*NOAA standard scale (Dlugokencky et al., 2005)*
*ALE/GAGE calibration procedure (Prinn et al., 2000)*

Observations used for model verification, lines 199-205: Why has only this very limited set of atmospheric stations been used (see general comment (1))?
*As mentioned above, now 16 NOAA stations*

Observations used for model verification, lines 205-207: "Monthly mean mixing - unfiltered with respect to local pollution events - are compared to respective monthly averaged model samples...": Why did the authors use monthly mean values, and not hourly or daily averages (see general comment (2)) ?
*As mentioned above, consistently with the measurements, the model now samples at every time step and averages at the end of every month (sub-model SCOUT).*
*In consideration of the 20 year's simulation period and the >8 years lifetime of methane are comparing monthly means.*
Observations used for model verification, lines 209ff: which calibration scale has been used for the CARIBIC CH4 measurements?
*We inserted:*
 *"The calibration is carried out using NOAA Methane WMO scale (Dlugokencky et al., 2005) For further information . . ."*

[Figure]

Simulation results, lines 229-230, "spin-up simulations and scaled to match the 1997 station measurements", and lines 254-255 " For initialization, a global methane distribution pattern for January was created iteratively in several and finally rescaled to Jan. 1997 station measurement data": The spin-up and scaling should be described in more detail (but best in section with model description): how long is the spin-up, which emissions have been used (probably the same as for the period 1997-2006?)? Did you just scale the calculated 3D fields?
*We introduced in*
**Ch. 2, Model Setup:**
*"Using a priori emission estimates, an initial CH$_4$ distribution was derived in the course of several spin-up simulations repeated until a steady state global CH$_4$ mass has settled over the years 1997 through 2006."*

**Ch. 2.2.1 Methane emissions:**
*"The GFED biomass burning statistics include agricultural waste burning events. Biomass burning emissions are inter-annually variable and the 1997 emission was 2.4 times as high as the 1998-2015 average (Fig. S1c)."*

**4.1 The period 1997 through 2006**

*"For initialization, a global methane distribution pattern for January was created as mentioned above and ensures a balanced annual average global $CH_4$ mass over the entire period with inter-annually constant sources and sinks up to deviations caused by variations in biomass burning."*

**Ch. 4.1.1 NOAA/AGAGE stations**

*"The initial distribution, which is the result of a long term simulation, does not precisely reflect the special Jan. 1997 situation, but obviously overestimated starting values at northern hemispheric stations level down in the course of the first six months. The Solver cannot improve this because it acts on the whole biomass burning series and not on individual years."*

If so, there would be some inconsistency between the applied emissions and the concentrations (which may also explain why the simulated CH4 concentrations still increase between 1997 and 2000).

*Figs. 1 and 5 in the new paper version (cf. Fig. 1 for CGO at the end of this document) reveal an increase also in the between 1997 and 2000. NH-enhanced values level down within six months and are caused by the anomaly in GFED biomass burning.*

Simulation results, lines 265-269: "The linear dependency between source strength and atmospheric abundance...", and lines 286-289: "The integrated model CH4 masses exactly match the mass calculated": this has already been discussed before.
*New formulation:*
*"The linear dependency between source strength and atmospheric abundance is reflected in the model's partial differential equation system and allows the redistribution . . ."*

Simulation results, line 310: "... fossil group of categories comprising landfill-, coal-, gas-, and oil (FOS)": CH4 from landfills are (usually) not fossil, but primarily from relatively recent carbon.
*In the new simulation that is no longer an issue: all categories are considered individually.*

Simulation results, line 346: "In contrast to the monthly average station data, the CARIBIC individual methane observations...": The station data - as provided to users - are hourly data. See also general comment (2).
*The usage of the sub-model SCOUT, consistently with the measurements, samples at every time step and averages at the end of every month.*

Simulation results, lines 365-366: "...suggesting that the vertical resolution of the model grid is not optimal to resolve the fine structure in the tropopause region.": Probably this is not only due to coarse vertical resolution, but also due the vertical CH4 gradient in the stratosphere.

*The comment of the referee is not in contradiction to our conclusion: The tropopause region with the declining vertical CH4 gradient cannot be properly resolved due to the course vertical resolution (~500m) of the hybrid model grid at this altitude.*

Simulation results, line 369: "according to the definition in Sect. 3.2 (Fig. S3)..." I assume this should be Fig. S2?
*According to the referees comment, in the updated version,*
*Fig. S3 now is correctly referred as Fig. 2b*

Simulation results, lines 386-387: "Figs. 12a and b...logarithmically scaled": the figures seem so use a linear scale. Furthermore, the figures show concentrations, while the figure caption states "Assumed additional emissions..." (Should be rephrased to e.g. "Impact of assumed additional emissions...").
*According to the referees comment Fig. 12 is now Fig. S2 in the updated version and log scaled.*
*We rephrased to "Impact of assumed additional emissions . . ."*

Simulation results, lines 391-392: "...upper estimate from Bergamaschi et al. (2013) of 22 Tg CH4 yr-1 as a first guess": It should be mentioned that the estimate of Bergamaschi et al. (2013) is for a different time period (2007-2010), compared to the 2007-2014 period used in this study.
*Following the referee's advice we avoid a citation because the 28 Tg CH4/y in the new version this has to be considered just as an upper limit for the Solver:*
*"We used an upper limit emission of 28 Tg CH$_4$/yr to be added in order to fit the upward trend."*

Simulation results, lines 395-396: "Both scenarios perfectly reproduce the observed CH4 trend...": I would suggest to avoid the term "perfectly".
*We avoided the term "perfectly".*

Simulation results, lines 400-401: "Changes in the removal rate of methane by the OH radical have not been seen in other tracers of atmospheric chemistry, e.g. methyl chloroform (CH3CCl3) (Montzka et al., 2011; Lelieveld et al. 2016) and do not appear to explain short-term variations in methane.": I do not agree with this statement. Although Montzka et al. [2011] derive only small interannual variability their CH3CCl3 based estimates still show variations on the order of +/- 3%, which is equivalent to a variability of the OH sink of +/- 17 Tg CH4 yr-1. Furthermore, the recent papers of and Turner et al. [2017] demonstrated the potential significant impact of variations in OH on the trend and inter-annual variability of CH4. I suggest to include the references to the two papers.
*Following the referee's advice we refer to Turner et al. (2017) in the new text. :*
*"Turner et al. (2017) based on numerical analyses find that a combination of decreasing methane emissions overlaid by a simultaneous reduction in OH concentration (the primary sink) could have caused the renewed growth in atmospheric methane. However they cannot exclude rising methane emissions under time invariant OH concentrations as a consistent solution to fit the (rising) observations."*

Simulation results, lines 423-424: "This shows that, when the SHA emissions are located away from the North America, no fraction is found that could minimize simultaneously the _NS and RMS": Given the very limited number of stations (see general comment (1)) and the question how well coastal / regional stations are simulated by the EMAC model (see general comment (2)), the question is, if this finding is really significant / robust.

*The new simulation now considers 16 stations– and optimization procedure upgraded*

Simulation results, line 437: "linear trend lines 0.32x (CARIBIC) and 0.31x (EMAC )":
units are missing

*New formulation:*

*"The slopes of the linear trend lines 0.32x (CARIBIC) and 0.31x (EMAC) over time where x = number of months . . ."*

Simulation results, lines 451-452: "the tropopause influence is stronger": probably also the influence of the lower stratosphere.

*New formulation:*

*"the influence of the lower stratosphere is stronger, leading to reduced linear slopes together with comparably less $R^2$ of 0.59 and 0.72 ( Fig. 14). "*

Conclusions and Outlook, lines 488-496: Would be useful to expand the conclusions, including a discussion / summary of the novel aspects of this study, the uncertainties of the results and limitations of the study. Furthermore, it should be summarized, how the results from this study compare with the existing literature studies.

*A co-author is taking care for this and his findings will be incorporated in the conclusions as soon as possible.*

Conclusions and Outlook, lines 497-499: "In view of the additional global CH4 source since 2007, a source – sink equilibrium has not yet established after the 8 years of emissions considered. A 2nd order polynomial extrapolation predicts steady state after 13 years, assuming that the emissions remain unchanged.": This scenario seems quite hypothetical, and global emissions (including their latitudinal distribution) remaining constant over 13 years relatively unlikely.

*Following the referee's advice  we included in*

**4.2 Simulating the recent methane trend** *- first paragraph:*

*"Between 2007 and 2013 the slope appears nearly linear (Fig. 1), and the discrepancy can be removed by assuming an additional constant $CH_4$ source for this period. We find that after 2013 an additional increment is necessary to fit the trend."*

*and in*

**5 Conclusions and Outlook:**

*"A 2nd order polynomial extrapolation predicts steady state after 13 years, assuming that the emissions remain unchanged, which does not seem realistic in view of the observed development after 2013/14 (Fig. 1). "*

Conclusions and Outlook, lines 497-499: "Nevertheless, the degree of freedom in the

choice of sources is limited,...": Taking into account also uncertainties in the spatial (and temporal) distribution of emissions, a very large number of emission scenarios is possible - while only 2 scenarios were investigated in this study (see gen. com. (3)).

*Following the referees comment we formulated:*

*"Two possible additional methane sources, shale gas extraction (SHA) and tropical wetlands (TRO) have been investigated, that could cause the resuming methane growth since 2007. We showed that a methane increase of 25.47 Tg/y in 2007 and subsequent years, of which 69 % from TRO and 20 % from SHA, can optimally explain the recent $CH_4$ trend until 2013."*

*As mentioned above, realizing a significant underestimation from 2007 on, just the two hypothetical scenarios TRO and SHA have been included for sensitivity testing of the observed trend.*

Table 1: references should be given for the individual a priori emission estimates.
*The methane emissions are based on (Houweling et al. 2006)*
*The "burning"-part of the GAMeS dataset is replaced by the GFEDv4s statistics (Randerson et al., 2018) in addition to biofuel combustion emissions from the EDGAR2.0 database (Olivier, 2001).*

Figures - general comment: The number of figures seems very large - several of them could be put in the supplementary material.
*Following the referees comment we reduced the number of figures.*

Figure 1: How were the data fitted?
*The data have been fit by linear regression – now formula inserted for better interpretation.*

Figure 3: Explain the meaning of the individual red circles
*Figure no longer used*

Figure 4a: "Zero" point (a priori emissions) should be indicated
*Figure no longer used because of non-linear Solver approximation in the new version.*

Figure 4b: What is the meaning of the curves (interpolation) between the individual stations?
*In the new version "Figure 3 at the end of this document" includes 16 stations and interpolating lines left out.*

Figure 5: "dashed line": The figures seem not to show any dashed line.
*"dashed line": Text updated.*

Figure 7: What is the meaning of the colors?
*now "Figure S6" no more color shift*

Figure 11b: Is the shown average for all CARIBC flights as function of time really very useful? Probably the spatial coverage of the flights is also changing significantly over time.
*It turned out to be useful, because in the beginning we were misguided by some Africa flights in a way that we tended to blame to low model mixing ratios to convection problems. This graph finally demonstrated the methane increase and the effect on Africa flights which started in 2009.*

Figure 13b: What is the meaning of the curves (interpolation) between the individual stations?
*In the new version "Figure 10" includes 16 stations - interpolating lines in this graph are helpful to visually associate the points to a category. (cf. Fig. 4 at the end of this document)*

Figure 16: (as for Figure 11b): How is the spatial coverage of the flights changing over time?
*The spatial coverage of the flights changing over time used to be depicted in Fig.S6, which is now Fig. S7 in the new paper version.*

Figure 17: Legend needs to be explained. Which curves are for which period?

*Fig. 17 is now Fig. 15 in the new version and the legend is updated (cf. Fig. 15copy at the end of this document).*

References

Dlugokencky, E. J., L. Bruhwiler, J. W. C. White, L. K. Emmons, P. C. Novelli, S. A. Montzka, K. A. Masarie, P. M. Lang, A. M. Crotwell, J. B. Miller, and L. V. Gatti, Observational constraints on recent increases in the atmospheric CH4 burden, Geophys. Res. Lett., 36(L18803), doi: 10.1029/2009GL039780, 2009.

Dlugokencky, E. J., E. G. Nisbet, R. Fisher, and D. Lowry, Global atmospheric methane: budget, changes and dangers, Phil. Trans. R. Soc. A 369(1943), 2058-2072, doi: 10.1098/rsta.2010.0341, 2011.

Rigby, M., R. G. Prinn, P. J. Fraser, P. G. Simmonds, R. L. Langenfelds, J. Huang, D. M. Cunnold, L. P. Steele, P. B. Krummel, R. F. Weiss, S. O'Doherty, P. K. Salameh, H. J. Wang, C. M. Harth, J. Mühle, and L. W. Porter, Renewed growth of atmospheric methane, Geophys. Res. Lett., 35(L22805), doi:10.1029/2008GL036037, 2008.

Rigby, M., S. A. Montzka, R. G. Prinn, J. W. C. White, D. Young, S. O'Doherty, M. F. Lunt, A. L. Ganesan, A. J. Manning, P. G. Simmonds, P. K. Salameh, C. M. Harth, J. Mühle, R. F. Weiss, P. J. Fraser, L. P. Steele, P. B. Krummel, A. McCulloch, and S. Park, Role of atmospheric oxidation in recent methane growth, Proceedings of the National Academy of Sciences, 114(21), 5373-5377, doi: 10.1073/pnas.1616426114, 2017.

Saunois, M., P. Bousquet, B. Poulter, et al., Variability and quasi-decadal changes in the methane budget over the period 2000–2012, Atmos. Chem. Phys., 17(18), 11135-11161, doi: 10.5194/acp-17-11135-2017, 2017.

Schaefer, H., S. E. Mikaloff Fletcher, C. Veidt, K. R. Lassey, G. W. Brailsford, T. M. Bromley, E. J. Dlugokencky, S. E. Michel, J. B. Miller, I. Levin, D. C. Lowe, R. J. Martin, B. H. Vaughn, and J. W. C. White, A 21st century shift from fossil-fuel to biogenic methane emissions indicated by 13CH4, Science, doi: 10.1126/science.aad2705, 2016.

Turner, A. J., C. Frankenberg, P. O.Wennberg, and D. J. Jacob, Ambiguity in the causes for decadal trends in atmospheric methane and hydroxyl, Proceedings of the National Academy of Sciences, 114(21), 5367-5372, doi: 10.1073/pnas.1616020114, 2017.

Worden, J. R., A. A. Bloom, S. Pandey, Z. Jiang, H. M. Worden, T. W. Walker, S. Houweling, and T. Röckmann, Reduced biomass burning emissions reconcile conflicting estimates of the post-2006 atmospheric methane budget, Nature Communications, 8(1), 2227, doi: 10.1038/s41467-017-02246-0, 2017.

[Figure]

*Figure 1:  The integrated annual amount was calculated 5.7e12 g higher by the solver.*

[Figure]

*Figure 2: AGAGE station CGO, Cape Grim, Tasmania:*
*Observations (blue) vs simulation (red) – $R^2$ = .93*

[Figure]

*Figure 3: Optimization of calculated ground station CH4 mixing-ratios towards observation (blue) in north-south direction by scaling the tagged emission contributions (now Fig. 6 in paper).*

[Figure]

*Figure 4: Scaling TRO and SHA emission fractions to fit the all-station observations within smallest RMS:*

*Left: Observations (blue) and total calculated CH4 without- (black), and with (red) trend period emissions (solid lines right panel).*

*Right: A-priori estimates (dashed) and solver-scaled (solid) TRO (19.44)- and SHA (5.74 Tg/y) emissions for trend years. (now Fig. 10 in paper).*

[Figure]

*Figure 15copy: Frequency spectrum of CARIBIC observed and EMAC simulated CH4-mixing-ratios separately plotted for the years 2000-2006 and 2007-2014.*

---

## Editor Decision (ED1)

The manuscript uses a large number of ground based and aircraft data to evaluate the EMAC methane simulations and to attribute the observed methane trends to changes in specific emission changes.

However, the manuscript lacks of clarity in terms of clear hypothesis, key message and associated uncertainties. The added value compared to earlier studies has to be clearly stated.

The abstract needs to be restructured: 1. Key question, 2. Use of EMAC and observations, 3. Assumptions made for modeling, 4. Key results and added value compared to earlier studies (to what the no-trend period is due and to what the trend-period). Also it misses some quantitative information on the findings, e.g. (lines 28-30) by how much fossil fuel is reduced and tropical wetlands and rice paddies are increased in the posteriori emissions?

Furthermore, there are a number of hypotheses made for this work that require to be clearly spelled out, justified and discussed with regard to the uncertainty they introduce to the conclusions of this study (as also pointed by the other reviewer). This has to be done early in the paper in section 2 where the model set up is explained. An appropriate place would be section 2.2., which could be 'Model set up and assumptions made'. These assumptions are mainly:

1.      the no-interannual variability of OH radical and thus the use of a prescribed OH radical concentration, which also implies no chemical feedbacks and linearity in the chemical destruction of CH4. The use of the prescribed OH is mentioned in the model set up while references that can partially support such choice are coming later in page 12 (lines 440-444). No discussion is really made on the uncertainty introduced in the results from such assumption.
2.      the constant interannual CH4 emissions for the a priori scenario
3.      testing different changes in specific sources  emission  intensity but assuming the constant geographic distribution per source.
4.      Neglecting a change in the soil sink? When optimizing for the emissions (or I misunderstood?)

The model description is also unclear at several places. For instance

1.      How the prescribed OH field has been derived (line 146 or later on line 212)?
2.      How the steady-state global CH4 simulation for the years 1997-2006 has been performed? (lines 152-153
3.      How many and which online simulations of EMAC have been performed ?
4.      And how many and which 'solver' fits have been performed?
5.      What kind of constrains are imposed to the 'Solver'
    a.      which constrains are exactly used in this study?
    b.      for which case studies is the Solver used? (A Table could be very illustrative for this).
    c.      As currently written, the reader has the impression that multiple (online) EMAC simulations have been performed and the figures presenting the model versus observations show the results of such simulations.

      d.      In lines 321-324 you discuss tolerance intervals; which are used for the various sources studied here?

6.      The tagged simulations (are online simulations?) have to be described in section 2 and not refer to them mainly in section 4 (results). Also give the simple equation (sum of tagged species) to explain how you calculate the total CH4 concentration.

In addition, there are several repetitions throughout the manuscript that have to be removed.

At several places text from other publications is used with quotation marks, for instance lines 82-92, or without, for instance lines 441-443. Such text has to be re-written in author's proper words with appropriate reference to the original study.

Furthermore, there are several long sentences difficult for the reader that can be broken down in 2 or 3 shorter ones and increase readability of the text (for instance lines 21-24).

More specific comments.

1.      Line: 13: what means 'in specified dynamics mode'? Is this comment needed?
2.      Lines 32- 35: '…added to the posteriori no-trend period emission distribution' rephrase.
3.      Lines 37-38: I suggest rephrasing starting by ' A combination of these sources that is statistically most likely excludes ….
4.      Line 39: This is not a sentence for the abstract. It can be used in the conclusions.
5.      Line 47: 'CH4 variability' do you mean 'CH4 mixing ratios'?
6.      Line 60: forcing of CH4 is… (0.6 Wm-2 instead of 0.5Wm-2)
7.      Line62: slowed down for about 8 years until sources and sinks quasi balanced
8.      Line 65: Please rephrase the last part of the sentence. It could be like the following: This reveals a period without-trend from 2000 through 2006 and one with increasing trend afterwards, which steepens after 2014.
9.      Line 73: 'the increase in CH4 since 2007' until when?
10.    Lines 96-103: Model evaluation is only one part of the study presented and is not enough for an ACP paper, although a very large dataset is used for this evaluation including the large number of samples collected during the CARIBIC flights. The suggested attribution of observed CH4 trends to potential emission changes seems an additional objective which increases the value of the study and has to be explicitly mentioned here.
11.    Lines 154-156 should move once the need for the use of the Solver is expressed i.e. line 158 before the beginning of the sentence.
12.    Line 162: 2.3.1 Methane a priori emissions
13.    Lines 181-183 contradict with lines 176-177, please rephrase to make it coherent.
14.    Lines 187-188 and 194-196 are results of this study (if I understand it correctly), why they are presented in the model -set-up section?
15.    Lines 200-202: Why the deposition velocity of CH4 would depend on CH4 mixing ratio? Do you mean deposition flux?

16. Line 208: 'photolysis and chemical reaction system' change to 'photochemical reactions system'
17. Line 214: global distribution of tagged methane or total?
18. Line 215 'model samples' this deserves a bit of explanation because is adding value to the comparison of model results with observations.
19. Line 251: from January 1997 (?) up to December 2016 … online model samples…
20. Line 254 'slowing increase' please rephrase
21. Line 262: For the period after 2007…
22. Line 268: what you mean with 'biogenic agricultural' ?
23. Lines 279- 281 – repetition with lines 256-260
24. Lines 284-287: move to data section 3.
25. Line 291: all source categories (?)
26. Line 298-299: repeated 3rd time here.
27. Line 309: refer to figure 3
28. Page 9: (1st and 3rd paragraph) Make clear how you calculate the interhemispheric difference and how much is in the a priori and how much is in the posteriori simulations.
29. Line 333:' all station' does this mean all (daily?) data from all studied surface stations? Please specify.
30. Lines 343-344. Please rephrase for clarity, as is appears wrong to me.
31. Line 355: do you mean 'assuming that the total sink if unchanged'
32. Line 364: CH4 appears to ..
33. Lines 370-371. Move the sentence before 'however' in line 372.
34. Line 374: upper troposphere
35. Lines 376-378: this is also true for the horizontal scale, please rephrase
36. Lines 375-376 and 381 have repetitions.
37. Lines 381-383: Please rephrase for clarity.
38. Line 404. Start new sentence with 'Turner et al.'
39. Line 414: 'same' with what?
40. Line 417: …performed with EMAC (?) with these sources…
41. Page 12: TRO,SHA, FAE better spell out – it is very hard to follow since the reader needs remember all these acronyms.
42. Lines 440-447: this argues in favor of neglecting the interannual variability of OH. Should be moved very early in the paper where presenting the overall assumptions made and arguing for them (see major comments). Also provide information on how much interannual variability is found in these studies.
43. Lines 458: 'additional a priori emissions' how they are derived?
44. Lines 460, 466: spell out ΔNH/SH
45. Lines 462: which are the four combinations and how they are chosen?
46. Lines 470-473: rephrase for clarity. 'Indications about the role of fossil…' to what?
47. Conclusions: here it is expected to present your finding with a critical view, i.e. provide also some information on the uncertainties associated with them.

I also recall the relevant comments of the last review on the earlier version of your manuscript, which seem not to be taken into account for the current revision and have to be addressed.

1-    Conclusion: " the presentation of the conclusions and outlook should better summarize the real conclusions (and limitations) of the study. E.g. the discussion of the "2nd order polynomial extrapolation predicts steady state after 13 years" seems rather hypothetical and the statement " NOAA/AGAGE station methane data are updated annually so further updates are expected" rather trivial." Please correct accordingly.

2 '"Enhanced precipitation in the regional summer season (Nisbet et al., 2016; Bergamaschi et al., 2013) may be a possible cause of growing tropical wetland emissions". The reviewer mentioned that: 'none of the 2 cited papers analyzed in any detail the tropical precipitation patterns, nor do Zimmermann et al. present any own meteorological analyses. While some studies in the literature found some correlations between tropical wetland CH4 emissions and ENSO induced anomalies in precipitation [e.g. Pandey at al., 2017], the reported anomalies appear directly related to the ENSO patterns (with a typical duration in the order of 1 year) - but to my knowledge do not support any persistent increase over the entire 2007-2016 period.'' So please rephrase that sentence accordingly (lines 407-408).

3. '"Kirschke et al. (2013) and Turner et al. (2016), however, found that an increase by 17-22 Tg/y could explain the renewed methane growth and 30-60% of this could be attributed to increasing U.S. anthropogenic methane emissions, which supports our results with 20.70 Tg/y emission increase including 8.38 Tg/y", but do not mention that the results of Turner et al. (2016) have been questioned by Bruhwiler et al [2017], highlighting in particular several methodical issues in the Turner et al. (2016) paper. In general the discussion of the results should be put much more in context with the existing literature."

I do not see how this point of the reviewer has been addressed in the current version. (lines 404-406).

4. Furthermore the reviewer states:

 'A further issue of the paper is that the applied optimization technique is rather simple. In principle such simple techniques (with a very limited number of parameters) can be useful for quick analyses and for illustrative purposes. However, this should be put into context with more sophisticated inverse modelling techniques and should include a discussion of the limitations of these simple techniques. The optimization technique used in this study is rather similar to simple synthesis inversion techniques, typically used ~20 year ago (e.g. [Hein et al., 1997]) - using fixed global spatial and temporal emission distribution patterns. A very critical issue of these synthesis inversions, however, is that they are prone to the so-called aggregation error [Kaminkski et al., 2001].'

I think that for clarity a comment in this direction is needed, i.e. that the applied optimization technique is rather simple, but can be useful for quick analyses and for illustrative purposes. However, we need to keep in mind that this type of inversions is prone to the so-called aggregation error [Kaminkski et al., 2001].

---

## Author Response (AR2)

*Peter Zimmermann, May 1, 2019*
*revised May 20, 2019*

**Reply to Re-Review of revised manuscript**

"Model simulations of atmospheric methane 1997-2016 and their evaluation using NOAA and AGAGE surface- and IAGOS-CARIBIC aircraft observations" by Zimmermann et al.

The revised manuscript of Zimmermann et al. presents an updated analysis of the global CH4 budget (and trends), using now 17 global monitoring stations instead of only 6 stations in the initial discussion paper, while leaving out coastal stations (used in the initial discussion paper), which cannot be well reproduced by the EMAC model.

*Reply ▶: This is not correct. The effect of synoptic variations at coastal stations can be well simulated by the EMAC model; however NOAA observations are filtered selecting baseline conditions. We also note that EMAC is a most extensively documented model and that the nudging applied provides very realistic results. It is simply not true that off- line models a priori perform better.*

Therefore the authors addressed the general comments (1) and (2) of my previous review.

However, the authors did not address at all my previous comment (3) and still analyze only one hypothesis for the global CH4 increase, namely the combined increase of CH4 emissions from shale gas in North America and from tropical wetlands.

*▶It has been our intention from the beginning to analyze the effect of two scenarios and their combination on the post 2006 methane increase phenomenon.*
*1. Increased tropical rainfall*
*2. Increased North American shale gas fracking*
*We concluded that a combination of both best explains the observed global distribution represented by NOAA stations and also the large set of CARIBIC flight observations.*
*We do not claim that this is the only possible explanation.*

Given the large number of hypotheses discussed in the literature, I find it very unsatisfactory that the analysis of the paper is still limited just to this single scenario.

*▶ Yes, the number of scenarios is large and for a critical reader they are even bewildering. We clearly deal with 2 source scenarios and their combination. We do not cover all of the large number of hypotheses treated in the literature, for instance we do NOT invoke scenarios in which OH undergoes changes. Our results belong to the best traceable results published so far and we do not desire to contribute to a bewildering number of scenarios. However, following the referee's reference to Schaefer et al. (2016) challenging the importance of tropical wetland emissions in context with the post-2006 $CH_4$-trend, we are going to extend the number of scenarios and initiated the simulation of increased agricultural emissions (ruminants and rice paddies).*

40    In addition, the authors do not even make any attempt to motivate the choice of this particular scenario.

*►First of all, we note the extensive treatment in the literature and at meetings of methane emissions from fracking and also from tropical wetlands.*

*Following the reviewer's comment we extended our motivation part in the introduction (bold*
45    *faced, below):*
*The resuming upward trend after 2007 (Dlugokencky et al., 2009; Rigby et al., 2008, IPCC 2014) is not fully understood: data analysis (Nisbet et al., 2016, Worden et al., 2017) and inverse modelling studies (Bergamaschi et al., 2013) indicate that global emissions since 2007 were about 15 to 25 Tg $CH_4$/y higher than in previous years, possibly caused by*
50    *increasing tropical wetland emissions and anthropogenic pollution in mid-latitudes of the northern hemisphere. A potentially growing source that was identified is hydraulic shale gas fracturing, for instance in Utah, where 6 to 12 % of the natural gas produced may locally leak to the atmosphere (Karion et al., 2013, Helmig et al. 2016). The increasing production of fossil fuels to some extend may explain the $CH_4$ trend; however, Schaefer at al. (2016) by*
55    *means of $^{13}C/\,^{12}C$ ($CH_4$) data and a box model concluded that fossil fuel related emissions are a minor contributor to the renewed methane increase, compared to agricultural emissions dominated by ruminants and rice . Simultaneously, "since 2007 $\delta^{13}C$-$CH_4$ [...] has shifted to significantly more negative values suggesting that the methane rise was dominated by increases in biogenic methane emissions, particularly in the tropics, for example, from*
60    *expansion of tropical wetlands in years with strongly positive rainfall anomalies or emissions from increased agricultural sources such as ruminants and rice paddies (Nisbet et al., 2016)."* **In addition, Schaefer et al. (2016) found that tropical wetland emissions, even relatively $^{13}C$-enriched, match the post-2006 perturbation not as well as emissions from rice cultivation and ($C_3$-fed) ruminants. However, this isotopic evidence against tropical**
65    **wetlands is not strong, they claim that a sustained source $^{13}C$-depletion from a 2000-2006 average to 2011 by ~12 Tg is "harder to reconcile" with tropical wetlands as with other biogenic emissions, such as agricultural ones.**

While there is some evidence from several studies that agricultural CH4 emissions from ruminants have increased (a scenario which unfortunately is not further analyzed in the
70    paper), a persistent increase of tropical wetland CH4 emissions over the whole 2007-2016 period remains very speculative. The authors only briefly state in section 4.2 (lines 399-400): "Enhanced precipitation in the regional summer season (Nisbet et al., 2016; Bergamaschi et al., 2013) may be a possible cause of growing tropical wetland emissions". However, none of the 2 cited papers analyzed in any detail the tropical precipitation patterns, nor do
75    Zimmermann et al. present any own meteorological analyses.

*►We agree with the reviewer that the citations are wrongly placed. Indeed Nisbet et al., (2016) and Bergamaschi et al., (2013) did not analyze the precipitation; on the other side both author suggested that the increased wetland emissions are due to the increased precipitation. Therefore the sentence has been reformulated:*

80    *``Enhanced precipitation in the regional summer season may be a possible cause of growing tropical wetland emissions (Nisbet et al., 2016; Bergamaschi et al., 2013)``*

While some studies in the literature found some correlations between tropical wetland CH4 emissions and ENSO induced anomalies in precipitation [e.g. Pandey at al., 2017], the reported anomalies appear directly related to the ENSO patterns (with a typical duration in the order of 1 year) - but to my knowledge do not support any persistent increase over the entire 2007-2016 period.

*►This could be subject of a follow up publication covering some more years beyond 2016. We do point out, that we cannot test all possible hypotheses in one paper.*

Given the restriction of the paper to a single scenario (which - in my view - is not the most likely one) the added value of the paper to the analysis of the global CH4 trends remains very limited.

*►The use of 2 source changes often mentioned in the literature, together with an extensively documented, well performing model, using surface data and a large aircraft data set, and applying basic statistical analysis makes our paper a useful contribution. Our results are throughout traceable and clear.*
*Nevertheless we are going extended the number of scenarios to 4.*

Furthermore, the presentation of the paper is not satisfactory - in the following just some examples:

(a) Introduction, page 3 (lines 84-86): "In other words, it was found by both authors that the combined fossil CH4 sources (1985-2002) must have been much stronger (factor of 2), at the expense of microbial sources." This conclusion was drawn only by Schwietzke et al. (2016), but not by Schaefer et al (2016). Then the authors continue: "Further, it was concluded that fossil fuel related sources had decreased." This was indeed concluded in both studies. However then the text continues: "Although the findings of the two articles are not necessarily in conflict...." As the studies have first been presented to give the same conclusions (which I think is not correct regarding the first statement): why should they then be in conflict?

*►Following the suggestion of the reviewer, the text has been modified as follows:*

*``As mentioned above, Schaefer et al. (2016) showed that "after 2006, the activation of biogenic emissions caused the renewed $CH_4$ rise", thus raising concern about the contribution from rice production versus wetland emissions. Further, it was concluded that fossil fuel related emission sources had decreased despite increased fossil fuel production and use owing to industry efficiency improvements (Schwietzke et al., 2016).``*

(b) Presentation of results: The first 1.5 pages of the results section ("4 Simulation results" and "4.1 The period 1997 through 2006" largely discuss again the model setup, especially the use of the tagged tracers (repeating 3 times the finding that the sum of the tagged methane tracers is equal to the "reference tracer" with all emission sources - a finding which is basically trivial, since the system is setup as linear system).

*►This was requested by the Co-editor.*

(c) apart from the restriction of the analysis to a single scenario (see my general comment above) the discussion of this single scenario is only very limited. At the end of section 4.2.1 the authors state:

"Kirschke et al. (2013) and Turner et al. (2016), however, found that an increase by 17-22 Tg/y could explain the renewed methane growth and 30-60% of this could be attributed to increasing U.S. anthropogenic methane emissions, which supports our results with 20.70 Tg/y emission increase including 8.38 Tg/y", but do not mention that the results of Turner et al. (2016) have been questioned by Bruhwiler et al [2017], highlighting in particular several methodical issues in the Turner et al. (2016) paper. In general the discussion of the results should be put much more in context with the existing literature.

►*The following text has been added, thanks to the referee's comment:*

``*[…] which supports our results with 20.7 Tg/y emission increase including 8.38 Tg/y (40 %) SHA. It must be however mentioned that Bruhwiler et al., (2017) found no emission trend for this source for the period 2006-2012, suggesting that previous calculations may be wrong due inherent difficulties in estimating emissions trends from column measurements*``

(d) the presentation of the conclusions and outlook should better summarize the real conclusions (and limitations) of the study. E.g. the discussion of the "2nd order polynomial extrapolation predicts steady state after 13 years" seems rather hypothetical and the statement " NOAA/AGAGE station methane data are updated annually so further updates are expected" rather trivial.

►*Following the referee's comments, the text on steady state after 13 years and on NOAA/AGAGE update were removed.*

A further issue of the paper is that the applied optimization technique is rather simple.

►*We are aware of this and it was our intention to present a simple and transparent method, which obviously works.*

In principle such simple techniques (with a very limited number of parameters) can be useful for quick analyses and for illustrative purposes. However, this should be put into context with more sophisticated inverse modelling techniques and should include a discussion of the limitations of these simple techniques.

►*The limitation is the suppression of feedback on the chemical reactors ($H_2O$, OH, Cl, and $O^1D$) which, however, is expected to be small and even negligible. Under this assumption – as mentioned several times - the model equation system is linear and there is no limitation inherent in the method we apply (except for the discretization, a common feature of all grid models). Furthermore, our EMAC system is not suitable for inverse modelling applications, for which other tools have been developed. Such tools are not necessarily more sophisticated but different.*

The optimization technique used in this study is rather similar to simple synthesis inversion techniques, typically used ~20 year ago (e.g. [Hein et al., 1997]) - using fixed global spatial and temporal emission distribution patterns. A very critical issue of these synthesis inversions, however, is that they are prone to the so-called aggregation error [Kaminkski et al., 2001].

►*We thank the referee for pointing this manuscript out. Nevertheless, this does not apply to our work. In fact Kaminkski et al.(2001) showed that this aggregation error appears when the state function is clustered. In our case, we are not dealing with clustered regions, but rather*

*we combine highly resolved global source segregated tracer distributions which are not subject to aggregation errors in the sense of Kaminkski.*

Given the discussed limitations of the revised paper, I cannot recommend the paper for
165    publication in ACP.

►*We do not understand why the reviewer responds so negatively to our work and the changes we have implemented subsequent to his/her review. Since we implemented two of the three main recommendations, and we had good arguments to not include the third, it seems*
170    *unfair that the reviewer subsequently downgraded the overall evaluation of our manuscript.*

[revised manuscript text omitted]

---

## Author Response (AR3)

Letter to the editor

Dear Editor, with this letter, the questions raised in the second round of review have been addressed. Here the comments are written in bold, followed by the reply. The overall text has been strongly revised, and therefore it was not feasible to add an annotated manuscript.

**REFEREE #1**

**The manuscript by Zimmermann et al. presents findings of a modeling study trying to identify the source strength of CH4 from different sectors, so that the ECHAM5/EMAC/MESSy modelling system properly reproduces the measured CH4 concentrations throughout a 20 year period. In the manuscript the model setup is explained and the results are presented, supported by an abundance of figures.**

We thank the referee for reading and comment the second version of our manuscript.

**There are a few minor points that I would like to see addressed before final publication on ACP:**

**1 - The authors should mention a reason why they are not using information for the inter-annual variability of CH4 emissions. It is clearly stated that inter-annually constant natural and anthropogenic emissions are used, when there have been regional - and global, changes within the period they are simulating (e.g. see "Global trends of methane emissions and their impacts on ozone concentrations" (doi:10.2760/820175)). The do give an explanation on why they do not use inter-annually changing OH concentrations, supported by references, but not for the emissions. If they choose not to include these changes there should be a short assessment explaining how they believe this affects the results.**

In our approach we adopted the geographical and seasonal distribution of a given set of emissions and assume their amounts to be constant over the "no-trend period" 1997 through 2006, which we see as permissible with regard to the long CH4 lifetime. We agree that this is an important assumption, and we underline this issue in the conclusions.

**2 - The sentence in lines 139-141 is repeated 10 lines later (149-151). One of the two should be changed/removed.**

The text has been completely rewritten for better readability.

**3 - P4 L152: The authors state that the model properly represented the years 1997-2006 after multiple spin-ups. Up to this point in the manuscript there is no mention of any kind of emission optimizing. This means that the model, with inter-annually constant emissions, without any emission optimization taking place, captured both the increasing concentrations of the years 1997-2000, and the stagnated period of 2000-2006. If this is the case, I would assume that the model was not yet in equilibrium, since there should be a linear relation between emissions and CH4 concentrations.**

We agree with the referee and we would like to mention that the non-equilibrium in 1997-200 is due to strong biomass burning event which does influence the following years due to the long methane lifetime (see for example Granier et al. (2011, doi: 0.1007/s10584-011-0154-1).

**EDITOR'S REVIEW**

**The manuscript uses a large number of ground based and aircraft data to evaluate the EMAC methane simulations and to attribute the observed methane trends to changes in specific emission changes.**

**However, the manuscript lacks of clarity in terms of clear hypothesis, key message and associated uncertainties. The added value compared to earlier studies has to be clearly stated.**

**The abstract needs to be restructured: 1. Key question, 2. Use of EMAC and observations, 3. Assumptions made for modeling, 4. Key results and added value compared to earlier studies (to what the no-trend period is due and to what the trend-period). Also it misses some quantitative information on the findings, e.g. (lines 28-30) by how much fossil fuel is reduced and tropical wetlands and rice paddies are increased in the posteriori emissions?**

Dear Editor, following the main comment we decided to rewrite the manuscript to make it more readable and easier to follow. Not only the abstract has been reformulated, and all the repetitions have been removed.

**Furthermore, there are a number of hypotheses made for this work that require to be clearly spelled out, justified and discussed with regard to the uncertainty they introduce to the conclusions of this study (as also pointed by the other reviewer). This has to be done early in the paper in section 2 where the model set up is explained. An appropriate place would be section 2.2., which could be 'Model set up and assumptions made'. These assumptions are mainly:**

**1. the no-interannual variability of OH radical and thus the use of a prescribed OH radical concentration, which also implies no chemical feedbacks and linearity in the chemical destruction of CH4. The use of the prescribed OH is mentioned in the model set up while references that can partially support such choice are coming later in page 12 (lines 440-444). No discussion is really made on the uncertainty introduced in the results from such assumption.**

**2. the constant interannual CH4 emissions for the a priori scenario**

**3. testing different changes in specific sources emission intensity but assuming the constant geographic distribution per source.**

**4. Neglecting a change in the soil sink? When optimizing for the emissions (or I misunderstood?)**

We have summarized these assumptions in the conclusion of the manuscript where we think it would be more appropriate.

**The model description is also unclear at several places. For instance**

**1. How the prescribed OH field has been derived (line 146 or later on line 212)?**

**2. How the steady-state global CH4 simulation for the years 1997-2006 has been performed? (lines 152-153**

**3. How many and which online simulations of EMAC have been performed ?**

**4. And how many and which 'solver' fits have been performed?**

**5. What kind of constrains are imposed to the 'Solver'**

All these informations have been added to the manuscript.

**a. which constrains are exactly used in this study?**
**b. for which case studies is the Solver used? (A Table could be very illustrative for this).**
**c. As currently written, the reader has the impression that multiple (online) EMAC simulations have been performed and the figures presenting the model versus observations show the results of such simulations.**
**d. In lines 321-324 you discuss tolerance intervals; which are used for the various sources studied here?**

All these points have been now clearly addressed in the manuscript.

**6. The tagged simulations (are online simulations?) have to be described in section 2 and not refer to them mainly in section 4 (results). Also give the simple equation (sum of tagged species) to explain how you calculate the total CH4 concentration.**

**In addition, there are several repetitions throughout the manuscript that have to be removed.**
**At several places text from other publications is used with quotation marks, for instance lines 82-92, or without, for instance lines 441-443. Such text has to be re-written in author's proper words with appropriate reference to the original study.**

Done and quotations have been removed.

**Furthermore, there are several long sentences difficult for the reader that can be broken down in 2 or 3 shorter ones and increase readability of the text (for instance lines 21-24).**

**As the entire text has been revised, we do not have a point by point answer to the specific comments. The main revisions of the text are:**

The abstract has been completely rewritten, highlighting the main conclusion of this work. While the introduction remains mostly untouched, the section on the numerical model (Sect.2) has been re-structured to give a better overview of the emissions and the numerical work performed in our study, with special focus on the tagging and error minimization procedure. While the observations description has not changed, the further sections (4 and 5) where drastically reduced, by removing all the redundant information and trying to be clear and concise. We hope in this way to have increased the manuscript readability and to facilitate the editor's work in this way.

---

## Author Response (AR4)

**Reply to the Editor:**

**Comments to the Author:**
**Thank you very much for the large effort you made to restructure the manuscript. This now reads nicely and is very informative. However, there are a few small improvements needed before this is published in ACP.**
**Some of them are mentioned by the reviewer.**
**A few more corrections are listed here-below. Please make all appropriate corrections to the manuscript and submit a revised track changes document that will go through editor review.**

We thank the editor. All the suggested corrections have been implemented in the revised manuscript.

**Reply to Referee #1:**

**The manuscript reads well and highlights the significance of the different CH4 emission sectors for explaining the increasing trends between 2007 and 2016. I have one comment and a few technical corrections that should be addressed before final publication to ACP.**

**Comments:**

**My main comment is on the period between 1997-2006 (section 4 of the manuscript). This is clearly a "spin-up" period for the model.**

> Our definition of "spin up" is actually the derivation of an initial CH4 distribution for Jan. 1st 1997. Three simulations of the period 1997-2006 were performed until the annual mean global CH4 mass reached equilibrium.

**It is stated in section 2 that sources and sinks of CH4 are kept constant, which means that inevitably the model would eventually reach a "stagnant" period for CH4. Stating that the model captures the measured values for this period, and especially the values between 1997-2000 is somewhat misleading. I agree that the period between 2000-2006 has to be evaluated and mentioned, to support that the model has reached equilibrium, but the earlier years are not representative of the ability of the model to represent measurements. My suggestion is to tread the period 1997-2006 as a spin up period, and focus on the remaining (2007-2016).**

> We partially agree with the referee. In fact, the period 1997-2006 is used to optimize the emissions sectors, and therefore we believe to be essential for the following period. We therefore prefer to keep the text organized as it is.

**P8L297: Why use the 2000-2005 RMS instead of the 1997-2006 RMS, or at least the stagnant period of 2000-2006?**

> We thank the referee for pointing that out. Indeed, the period 2000-2006 was used and in the text a typo was present

All the technical corrections have been implemented in the manuscript

[revised manuscript text omitted]

b: same for CARIBIC flightsflight observations (blue dots) and compared to respective EMAC simulationsimulations without trend emissions (red dots). Superimposed), units in [nmol/mol]. Super-imposed lines represent respective 100 times sliding means for better visibility.

[Figure]

Figure 10: Calculated total CH$_4$ without- (black crosses), and with optimized trend period emissions (red dots). By scaling RIC, SHA, and TRO emission fractions, the station observations (blue circles) are approximated with smallest RMS . After 2013, the trend accelerates and additional emission assumptions are necessary.

[Figure]

Figure 11: Regression analysis of EMAC calculations vs. observations of CH$_4$ at NOAA stations ALT, RPB, and SPO for the trend years 2007 through 2016.

815

[Figure]

Figure 12:  2007 through 2016 CH₄ development at NOAA and AGAGE stations:
Observations (blue) vs. optimized RIC+TRO (2007 – 2013) emission incrementEMAC simulations (red).),
Period: 2007 through 2016, units: mixing-ratio in [nmol/mol]. The trend emissions are optimized for the linear rising
period 2007 through 2013.

[Figure]

Figure 13: Monthly averaged EMAC-CH₄, including trend and CARIBIC-2 observations from 2007 through 2014 for all data obtained from CARIBIC Whole Air Samples (WAS) in blue, and model results in red.

[Figure]

825

Figure 14: Linear regression between CARIBIC-2 samples and EMAC calculations for all trend period flights (2007 – 2014)  and for flight regions with more than 300 samples.

[Figure]

Figure 15: Frequency spectrum of CARIBIC observed and EMAC simulated CH$_4$-mixing-ratios separately plotted for

the years 2000-2006 and 2007-2014.

[Figure]

Figure 16:

EMAC CH$_4$ calculations (red) and CARIBIC-2 observations (blue) from 2007 through 2014 – all flight samples.

[Figure]

Figure 17:

a: CH$_4$ mixing ratios observed by CARIBIC (blue dashed, right axis) and calculated by EMAC (red dashed thick) and tagged rice related CH$_4$ (green, left axis) - India flights Aug. 2008. The thin red dashed line marks the simulation without trend period increment for reference.

850

855

b: CH$_4$ mixing ratios [nmol/mol] observed by CARIBIC (blue) during all India flights 1997 through 2012 and corresponding EMAC simulations (red). The large scatter requires the sliding average of 7 points (solid lines).

860